bioengineering/fluid mechanics

metachronal paddling, rowing, crustacean swimming, drag-based propulsion, aquatic locomotion

**Author for correspondence:**
Arvind Santhanakrishnan
e-mail: askrish@okstate.edu

# Hydrodynamics of metachronal paddling: effects of varying Reynolds number and phase lag

Mitchell P. Ford, Hong Kuan Lai, Milad Samaee and Arvind Santhanakrishnan

School of Mechanical and Aerospace Engineering, Oklahoma State University, Stillwater, OK 74078, USA

AS, 0000-0003-1800-8361

Negatively buoyant freely swimming crustaceans such as krill must generate downward momentum in order to maintain their position in the water column. These animals use a drag-based propulsion strategy, where pairs of closely spaced swimming limbs are oscillated rhythmically from the tail to head. Each pair is oscillated with a phase delay relative to the neighbouring pair, resulting in a metachronal wave travelling in the direction of animal motion. It remains unclear how oscillations of limbs in the horizontal plane can generate vertical momentum. Using particle image velocimetry measurements on a robotic model, we observed that metachronal paddling with non-zero phase lag created geometries of adjacent paddles that promote the formation of counter-rotating vortices. The interaction of these vortices resulted in generating large-scale angled downward jets. Increasing phase lag resulted in more vertical orientation of the jet, and phase lags in the range used by Antarctic krill produced the most total momentum. Synchronous paddling produced lower total momentum when compared with metachronal paddling. Lowering Reynolds number by an order of magnitude below the range of adult krill (250–1000) showed diminished downward propagation of the jet and lower vertical momentum. Our findings show that metachronal paddling is capable of producing flows that can generate both lift (vertical) and thrust (horizontal) forces needed for fast forward swimming and hovering.

## 1. Introduction

Aquatic crustaceans such as copepods, krill and mysids encompass one of the largest member groups of zooplankton [1–3], with tremendous diversity in sizes ranging of the orders of 0.1–100 mm.

Freely swimming crustaceans use a unique drag-based strategy for locomotion, in which closely spaced swimming limbs (pleopods) are rhythmically paddled in an adlocomotory sequence starting from the tail and progressing to the head. Each appendage is phase-shifted in time relative to the neighbouring appendage, resulting in a metachronal wave travelling along the same direction as the animal. In contrast with lift-based aquatic propulsion in fishes [4], and jetting propulsion in jellyfish [5] and squids [6], drag-based metachronal swimming has received limited attention in the literature [7–9]. From an ecological perspective, aquatic crustaceans form a crucial connection in planktonic food webs by grazing on smaller phytoplankton and serving as prey for larger, commercially important animals such as fishes [1]. Studies of the design, kinematics and hydrodynamics of metachronal propulsion can improve our understanding of crustacean foraging and ecologically important behaviour such as schooling. From an engineering standpoint, studies of metachronal swimming can guide the design and development of miniaturized underwater drones.

The hydrodynamic outcomes of metachronal propulsion depend on interactions between pleopod morphology and stroke kinematics. Flow generated by metachronal paddling has been investigated in several crustaceans [9–21]. In general, metachronal paddling has been observed to generate a caudoventrally angled jet [11,14]. The effect of the beating pleopods has been hypothesized [9] to provide weight support via lift-generation at lower speeds. In terms of swimming performance, Alben et al. [7] mathematically modelled the metachronal stroke of fast forward swimming euphausiids in comparison to other stroking patterns and found that the metachronal pattern gave the fastest average body speed. Synergistic interactions between neighbouring appendages were proposed to lead to this outcome. However, flow between neighbouring limbs was not modelled. A computational study by Zhang et al. [22] showed that higher volumetric flux was pushed towards the tail in metachronal motion as opposed to synchronous motion, but did not report the large-scale angled jets observed in live crustaceans [9,11,13,14]. A recent study [23] using mathematical models of two or more rigid (non-hinged) paddles showed that metachrony can effectively move flow, but inter-pleopod interactions and downward-directed jets were not reported.

A central limitation in existing studies of metachronal swimming is that they do not explain how unsteady flow interactions between neighbouring pleopods can lead to the development of caudoventral jets observed in organismal studies of hovering and forward swimming krill [9,11,13,14]. The primary goal of this study is to elucidate how introduction of phase lag in periodic oscillations of closely spaced hinged paddles (idealized pleopods) can generate continuous propulsive jets. We also examine flow generated by metachronal paddling under varying Reynolds number ($Re$), to understand how flow characteristics could change with increasing body size when stroke kinematics remain unchanged. A dynamically scaled robotic platform was developed for this study, fitted with scaled-up physical models of two-dimensional (2D) flat plate paddles. Hinges were included in the paddles to closely mimic pleopod unfolding in power stroke (PS) and folding in recovery stroke (RS) [8]. Two-dimensional particle image velocimetry (PIV) measurements were conducted along the central plane of the paddles to resolve flows generated by interactions of a pair of paddles and by four paddles. $Re$ was varied from 50 to 800 by changing viscosity of the fluid medium. Our findings show that phase lag between adjacent paddles plays a central role in generating a large-scale propulsive jet. This large-scale flow pattern results from the interaction of opposite-signed shear regions generated by metachronal motion of neighbouring paddles. Increasing $Re$ resulted in increasing vertical and horizontal momentum of the fluid flow and allowed the jet to propagate farther from the body, probably due to reduced viscous dissipation.

# 2. Experimental methods

This study used a dynamically scaled robotic platform mimicking metachronal paddling commonly seen in freely swimming crustaceans. The platform was fitted with scaled-up models of hinged paddles (idealized flat plate representations of crustacean pleopods), with length and inter-pleopod spacing approximately eight times greater than in Pacific krill [8,13]. Flow characteristics including velocity fields, vorticity and momentum were determined using PIV along the mid-plane of the paddles.

## 2.1. Robotic platform

The base of the paddling robot was 3D printed on a CraftBot 3D printer (Craft Unique, Stillwater, OK, USA) using PLA filament. The paddles measured 152.4 × 76 × 2.5 mm (width × height × thickness), and were hinged 38 mm from the tip (halfway down). Each paddle nearly occupied the entire available

width of the aquarium, in order to achieve a 2D idealization of crustacean pleopods. The ratio of the inter-paddle distance to paddle length on the model was 0.7, similar to Pacific krill [8]. Paddles were constructed from clear acrylic, which allowed laser light to pass. Paddles were mounted to 6.35 mm diameter aluminium shafts, which were mounted to the 3D printed base. Rotational motion was controlled using timing belts that were driven by two-phase hybrid stepper motors with integrated encoders (ST234E, National Instruments Corporation, Austin, TX, USA), with 20 000 steps per revolution resolution. The 6.35 mm diameter aluminium shafts protruded from either side of the 3D printed base, and were driven by the rotation of the timing belts. In order to mount timing belt pulleys to the shafts, 3.4 cm gaps were left between the paddles and the walls on either side of the robotic platform. A multi-axis motion controller (PCI-7350, National Instruments Corporation) and a stepper motor drive (SMD-7611, National Instruments Corporation) were used to control the stepper motors. Front and top views of the robotic model are shown in figure 1a,b, respectively. The root of the paddle was located 457 mm from the bottom of the tank, 25 mm below the free surface of the fluid. Acrylic sheets were attached to the head and tail ends of the platform, to allow for a no-slip condition at the top surface of the $775 \times 220 \times 403$ mm (length × width × height) aquarium and a far-field condition at the bottom. A false acrylic wall of 0.6 cm in thickness was placed on one side of the platform to reduce the width of the tank from 29 to 22 cm.

Kinematics were controlled via a LabVIEW program created using NI Motion Assistant software (National Instruments Corporation). For each experiment, stroke amplitude (SA) was $\pi/2$ radians, and stroke frequency ($f$) was 1.5 Hz. The time-varying position of each paddle was prescribed as a pure harmonic function given by the below equation

$$\alpha(t) = \frac{\pi}{2} - \frac{\pi}{4} \cos\left(2\pi\left[\frac{t}{T}\right] - \phi\right), \tag{2.1}$$

where $\alpha(t)$ is the instantaneous prescribed angle relative to horizontal, $t$ is time in seconds, $\phi$ is the phase lag in radians and $T$ is the stroke period ($T = 1/f = 0.667$ s). Phase lags between paddles were introduced to create metachronal waves. Phase lags (P4–P3, P3–P2 and P2–P1) used in this study were 0, $\pi/3$ radians (16.7% of cycle), $\pi/2$ radians (25% of cycle) and $2\pi/3$ radians (33.3% of cycle). P1 refers to the head-most paddle, while P4 refers to the tail-most paddle (figure 2). The metachronal stroking sequence begins at P4, so that the metachronal wave travels from P4 to P1 (tail to head). Each paddle was prescribed the same $\phi$ relative to the paddle immediately behind. Note that $2\pi$ radians corresponds to one stroke cycle, so phase lag $\phi$ can be specified in terms of percentage of cycle by dividing its value in radians by $2\pi$. The prescribed kinematics for each paddle are shown in figure 2 (solid lines) for each phase lag. PS was defined as paddle motion from the head to tail, and RS as motion from the tail to head (indicated in the inset of figure 2a).

## 2.2. Paddling kinematics

Kinematics achieved by the four paddles were tracked from raw PIV images using the image analysis program ImageJ [24]. The paddle angles ($\alpha(t)$, figure 2) were measured at 10 points in a cycle. The definition of $\alpha$ was identical to that used in a study of krill swimming kinematics by Murphy *et al.* [8]. $\alpha$ increases in PS (cycle fraction from 0 to 0.5) and decreases in RS (cycle fraction from 0.5 to 1). The results show that achieved paddle angles (markers in figure 2) closely follow the kinematics prescribed to the stepper motors (solid lines in figure 2) across all $Re$ and phase lag conditions.

The hinges on the paddles were allowed to rotate freely about angle $\beta$ (defined in figure 2a, same as in [8]). $\beta$ varied between a minimum angle of approximately 120° and a maximum angle of 180°. Variation of $\beta$ angles are provided as electronic supplementary material (figures S1–S3). $\beta$ reaches its maximum value for each paddle at the beginning of their respective PS as $\alpha$ accelerates during stroke duration from 0 to 0.25; and begins decreasing once $\alpha$ begins to decelerate during stroke duration from 0.25 to 0.5. The rate of decrease in $\beta$ increases with increasing $Re$, and is also delayed in time. In general, $\beta$ spends more time near its peak value than its minimum, which agrees with observations in hovering *Euphausia superba* [8].

## 2.3. Test conditions

A robotic platform was constructed in order to systematically examine the flow generated by the coordinated oscillation of multiple paddles under varying phase lag. To this end, the model was dynamically scaled to have paddle-based $Re$ and inter-paddle gap to length ratio similar to

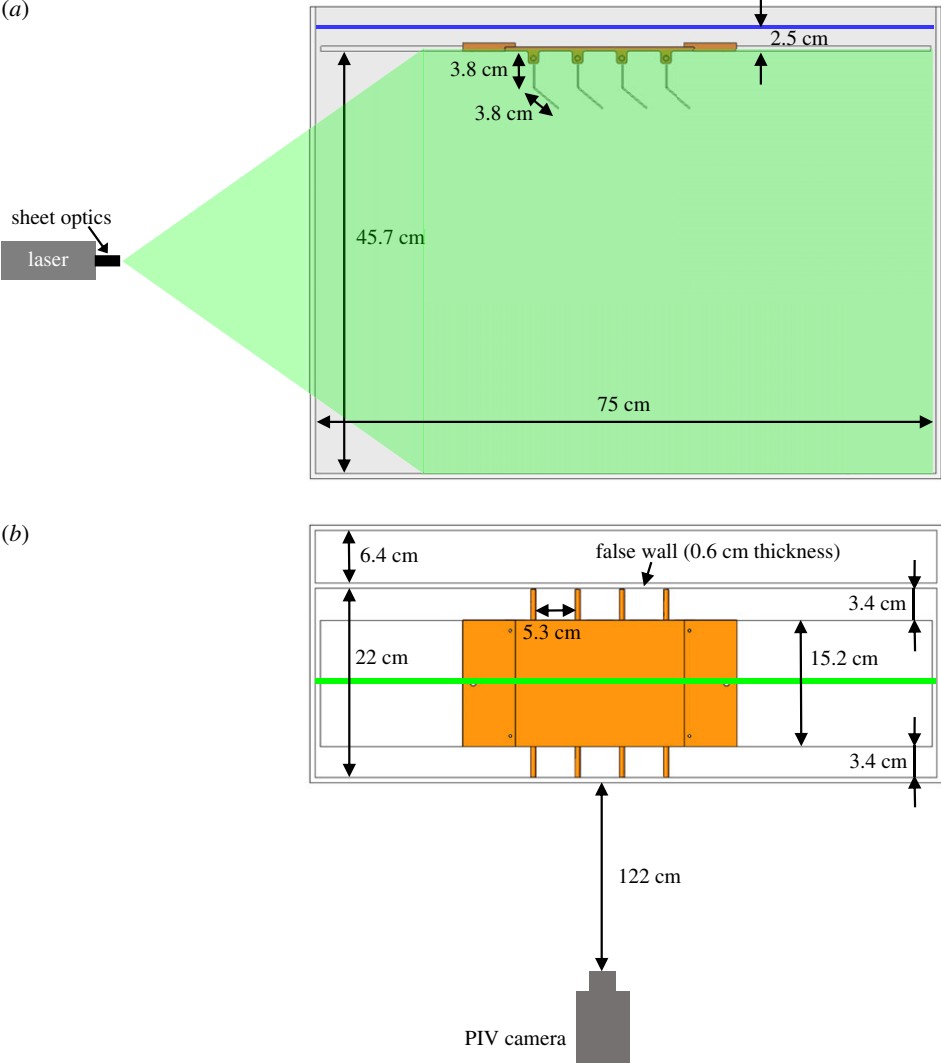

**Figure 1.** Dynamically scaled model and experimental set-up. (*a*) Front view of the robotic model showing the laser sheet used for 2D PIV measurements of synchronous and metachronal paddling. Each paddle physical model (2D idealized representation of crustacean pleopod) consisted of two thin acrylic sheets connected via a mechanical hinge. Arrangements of two paddles and four paddles were considered for this study, both with inter-paddle gap to paddle length ratio of 0.7 observed in Pacific krill [14]. Acrylic sheets were mounted to the model to ensure a no-slip boundary condition on the top surface of the fluid, while the depth of the aquarium allowed for a far-field condition near the bottom. (*b*) Top view of the paddling model showing the location of PIV camera. Laser sheet for PIV was located at the central plane of the paddles. Aluminium shafts (22 cm long, 6.35 mm diameter) protruding from either side of the model (top and bottom of (*b*)) were coupled to stepper motors with timing belts in order to drive each paddle with independently prescribed motion profiles. In order to mount timing belt pulleys to the shafts, 3.4 cm gaps were left between the paddles and the walls on either side of the robotic platform. These gaps have little impact on the flow at $Re = 50$, but do contribute to some out-of-plane flow at $Re = 250$ and $Re = 800$. A false acrylic wall (thickness = 0.6 cm) was used to reduce the width of the tank from 29 to 22 cm.

Pacific krill [8]. $Re$ was defined based on the maximum paddle tip velocity ($U_{\text{tip,max}}$), fully extended paddle length ($L = 76$ mm) and kinematic viscosity of the fluid ($\nu$) as

$$Re = \frac{L\, U_{\text{tip,max}}}{\nu}. \tag{2.2}$$

$U_{\text{tip,max}}$ occurs when a paddle is rotated by $\pi/4$ radians (=SA/2) from its initial position

$$U_{\text{tip,max}} = 2\pi f\left(\frac{\text{SA}}{2}\right)L. \tag{2.3}$$

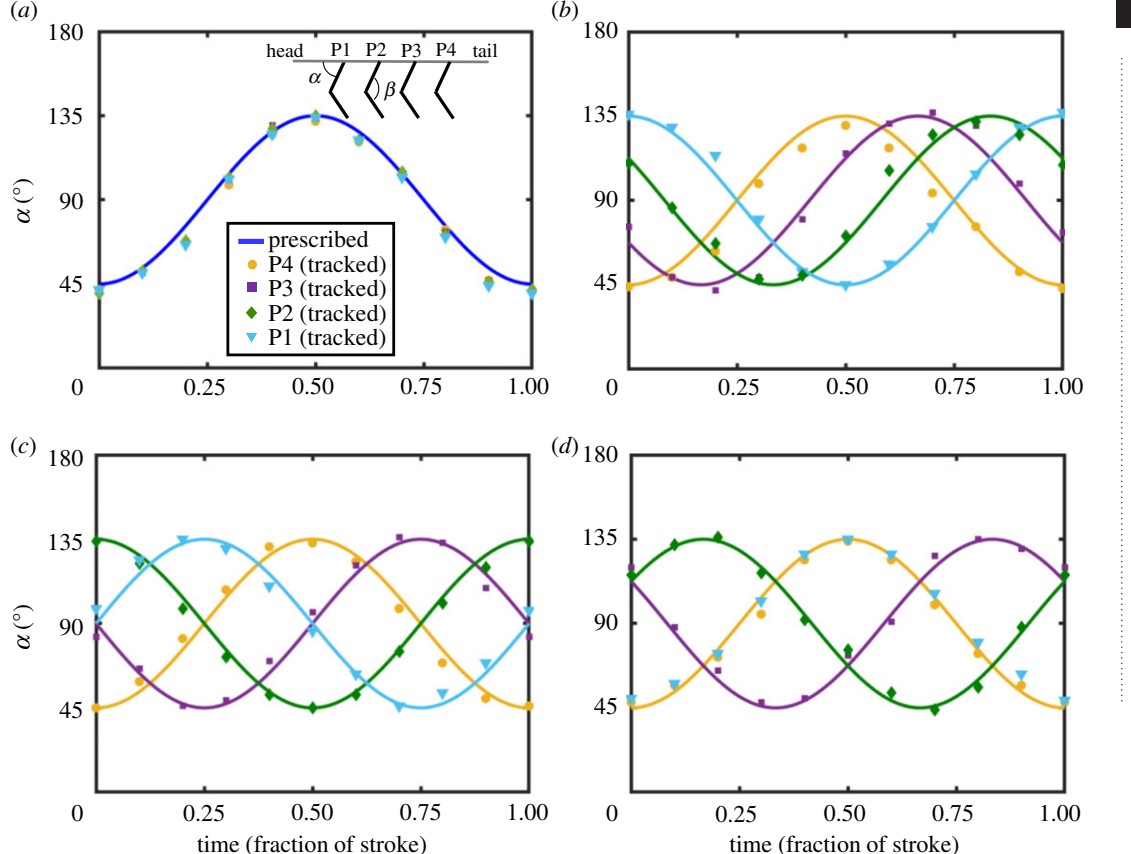

**Figure 2.** Prescribed and achieved motion profiles within a complete paddling cycle for: (*a*) 0% phase lag (synchronous paddling), (*b*) 16.7% phase lag, (*c*) 25% phase lag and (*d*) 33% phase lag. The time scale is non-dimensional fraction of stroke period ($T$ = 0.667 s). Schematic diagram in the inset of (*a*) shows: paddle angle $\alpha$; hinge angle $\beta$; and the head and tail ends of the acrylic sheets mounted to the model. P1 (left) is the head-most paddle, while P4 (right) is the tail-most paddle. $\alpha$ was prescribed as a periodic function given by equation (2.1). PS is defined as paddle motion in the head-to-tail direction, and RS is defined as paddle motion in the tail-to-head direction. Bulk flow is in the head-to-tail direction. Solid lines represent the prescribed angular position of the paddles over one cycle, whereas filled markers represent the angular position achieved by the robotic model.

Substituting $U_{\text{tip,max}}$ and SA = $\pi/2$ in equation (2.2) above results in the following relation for *Re*:

$$Re = \frac{\pi^2 f L^2}{2v}. \tag{2.4}$$

The definition of *Re* in equation (2.4) is based on the study by Zhang *et al*. [22], where it was estimated that *Re* of adult krill was roughly 600, *Re* of juvenile crayfish was of the order of 10 and *Re* of adult crayfish was between 250 and 1000. We tested *Re* = 50, 250 and 800 in this study, similar to the *Re* used by Zhang *et al*. [22] as well as the *Re* = 500 ± 300 observed in Pacific krill by Catton *et al*. [13]. To achieve these *Re*, *v* was varied using water–glycerin mixtures of different concentrations, according to the values shown in table 1, and measured using the Cannon-Fenske capillary viscometers (Cannon Instrument Company, State College, PA, USA). This resulted in the fluid density ($\rho$) having slightly different values for different *Re* cases, which was taken into account for momentum and momentum flux calculations. Varying phase lag did not affect *Re*, since stroke frequency (*f*) was constant.

Arrangements of two and four paddles were considered for this study. Inter-paddle hydrodynamic interactions were examined at higher PIV spatial resolution in the case of two paddles. Freely swimming crustaceans have four (e.g. crayfish [22]) or more pairs (e.g. krill [8], mysids [10]) of pleopods. The arrangement of four paddles was used to closely mimic the biologically relevant case. The experimental set-up was limited to four paddles on account of the control program that allowed prescribing only three independent motion profiles. Paddles P4, P3 and P2 were prescribed independent motion profiles, while P1 was electronically geared to move in the same (0, 33.3% phase lag) or opposite (16.7, 25%) direction of P4.

**Table 1.** Experimental conditions examined in this study. Paddle geometry and kinematics were maintained constant throughout this study. The composition of the fluid medium (water–glycerin mixture) was changed in order to vary Reynolds number (*Re*), with values of density and kinematic viscosity as shown. Laser pulse separation interval ($\Delta t$) for 2D PIV measurements was chosen to provide sufficient particle displacement for cross-correlation analyses.

| *Re* | density (kg m$^{-3}$) | kinematic viscosity (mm$^2$ s$^{-1}$) | $\Delta t$ (µs) |
|------|------|------|------|
| 50 | 1260 | 860 | 3500 |
| 250 | 1235 | 172 | 2750 |
| 800 | 1211 | 53 | 2000 |

## 2.4. Two-dimensional particle image velocimetry

Two-dimensional frame-straddling PIV was performed to visualize flow structures generated by metachronal paddling, and to quantitatively determine the flow field momentum at each *Re* and phase lag. The fluid was seeded with 55 µm diameter polyamide particles (density: 1.2 g cm$^{-3}$; LaVision GmbH, Göttingen, Germany). A 3–4 mm thick vertical laser sheet was generated using a −20 mm focal length plano-concave cylindrical lens and a double-pulsed Nd:YAG laser (Gemini 200-15, New Wave Research, Fremont, CA, USA) with 532 nm wavelength and maximum repetition rate of 15 Hz. This allowed the PIV system to capture frame pairs at 10 image pairs per stroke cycle. Images were captured using an sCMOS camera with the spatial resolution of 2560 × 2160 pixels and pixel size of 6.5 × 6.5 µm (LaVision GmbH, Göttingen, Germany). A 50 mm constant focal length lens (Nikon Micro Nikkor, Nikon Corporation, Tokyo, Japan) was attached to the sCMOS camera with the aperture set to 2.8 for all PIV measurements. The front of the lens was positioned 1.22 m from the front of the aquarium for the four paddle case, and was positioned closer for the two paddle case. An electronic trigger was generated at the start of paddling motion using the same LabVIEW program that controlled the stepper motors. PIV recordings were initiated after 100 stroke cycles were completed, to establish a periodic steady-state flow. For each test condition, image pairs were acquired at 15 image pairs per second ($f_{PIV} = 15$ image pairs s$^{-1}$) so that one image pair was obtained every 10% of stroke cycle. Image pairs were acquired across 30 consecutive stroke cycles ($N = 30$), for a total of 300 image pairs for each test condition. Laser pulse separation intervals ($\Delta t$) were changed for each *Re* to obtain maximum eight pixels displacement between two images of an image pair (table 1).

Multi-pass cross-correlation of image pairs was performed in DaVis 8.3 (LaVision GmbH, Göttingen, Germany), with one pass each of 64 × 64 pixels and 32 × 32 pixels with 50% overlap. Post-processing was performed to remove velocity vectors with peak ratio $Q < 1.2$, and 2D instantaneous velocity field data were exported from DaVis. Instantaneous velocity vector fields at each time-point $t$ were averaged across $N = 30$ consecutive stroke cycles to obtain a time-varying, *phase-averaged* velocity vector field

$$u(x,y,t) = \frac{1}{N}\left[\sum_{i=1}^{N} u_{inst}^{i}(x,y,t)\right]; \;\; v(x,y,t) = \frac{1}{N}\left[\sum_{i=1}^{N} v_{inst}^{i}(x,y,t)\right], \tag{2.5}$$

where $u_{inst}$ and $v_{inst}$ represent instantaneous velocity components in the horizontal ($x$) and vertical ($y$) directions, respectively, at time-point $t$. This operation resulted in 10 phase-averaged velocity vector fields per test condition with components ($u,v$). Phase-averaged velocity vector fields were further analysed to calculate $z$-vorticity (two and four paddles) and time-varying 2D components of linear momentum per unit width of the paddle (four paddles).

In addition to phase-averaging, time-averaging was performed across 300 instantaneous velocity vector fields to obtain a single *cycle-averaged* velocity field per test condition

$$\bar{u}(x,y) = \frac{1}{(Nf_{PIV}/f)}\sum_{i=1}^{Nf_{PIV}/f} u_{inst}^{i}(x,y,t); \;\; \bar{v}(x,y) = \frac{1}{(Nf_{PIV}/f)}\sum_{i=1}^{Nf_{PIV}/f} v_{inst}^{i}(x,y,t), \tag{2.6}$$

where $\bar{u}$ and $\bar{v}$ represent cycle-averaged velocity components in the horizontal ($x$) and vertical ($y$) directions, respectively. Cycle-averaged velocity fields were used to calculate horizontal and vertical momentum fluxes, magnitude of the linear momentum vector and angular orientation of the momentum vector generated in the four paddle case.

## 2.5. Definitions of calculated quantities

Vorticity was used to quantify fluid rotation in the flow field. The out-of-plane component of the vorticity vector ($z$-vorticity, $\omega_z$) was calculated in Tecplot 360 (Tecplot, Inc., Belleview, WA, USA) from phase-averaged velocity fields using the below equation

$$\omega_z = \frac{\partial v}{\partial x} - \frac{\partial u}{\partial y}. \tag{2.7}$$

Flow field momentum and momentum fluxes were calculated using custom Matlab scripts (The Mathworks, Inc., Natick, MA, USA). Phase-averaged linear momentum of a discrete fluid element (per unit width of the paddle) was calculated using the below equations

$$p_x(x,y,t) = \frac{1}{N} \sum_{i=1}^{N} [\rho u_{\text{inst}}^i(x,y,t) \, dx \, dy] \tag{2.8}$$

and

$$p_y(x,y,t) = \frac{1}{N} \sum_{i=1}^{N} [\rho v_{\text{inst}}^i(x,y,t) \, dx \, dy], \tag{2.9}$$

where $dx$ and $dy$ are the length and height of a fluid element, and $p_x$ and $p_y$ are the horizontal and vertical components of momentum per unit width of the paddle. Time-varying total momentum within the PIV field of view (FOV) was calculated using the below equations

$$p_x(t) = \frac{\rho}{N} \sum_{i=1}^{N} \left[ \iint u_{\text{inst}}^i(x,y,t) \, dx \, dy \right] \tag{2.10}$$

and

$$p_y(t) = \frac{\rho}{N} \sum_{i=1}^{N} \left[ \iint v_{\text{inst}}^i(x,y,t) \, dx \, dy \right]. \tag{2.11}$$

Cycle-averaged momentum flux across a line drawn in the FOV (figure 11$a$–$c$) provides an estimate of cycle-averaged force acting at that location. Cycle-averaged momentum fluxes per unit width were calculated using the below equations (2.12) and (2.13)

$$\text{HMF} = \rho \int |\bar{u}(x,y)| \, (\bar{U} \cdot \hat{n}) \, dy \tag{2.12}$$

and

$$\text{VMF} = \rho \int |\bar{v}(x,y)| \, (\bar{U} \cdot \hat{n}) \, dx, \tag{2.13}$$

where horizontal momentum flux (HMF) is the horizontal component of cycle-averaged momentum flux across a vertical line located at $x$, vertical momentum flux (VMF) is the vertical component of cycle-averaged momentum flux across a horizontal line located at $y$ and $\bar{U}$ is the 2D cycle-averaged velocity vector ($\bar{U} = \bar{u}\hat{i} + \bar{v}\hat{j}$). $\hat{n}$ is a unit normal vector from the line of interest, defined such that: $\hat{n} = \hat{i}$ for HMF (positive $x$; left to right) and $\hat{n} = \hat{j}$ for VMF (positive $y$; top to bottom). The dot product ($\bar{U} \cdot \hat{n}$) accounts for both magnitude and sign of the local velocity vector in the direction of $\hat{n}$. Multiplying this product with the absolute value of the velocity component ($|\bar{u}|$ and $|\bar{v}|$ in equations (2.12) and (2.13)) ensures that the integrand retains the sign. Positive values of HMF and VMF indicate momentum flux in the same direction as the corresponding unit normal vectors. The spatial limits of integration in $x$ and/or $y$ (equations (2.10)–(2.13)) cover the entire FOV in the horizontal and vertical directions, respectively.

Cycle-averaged total momentum was calculated using the below equations

$$\overline{p_x} = \frac{\rho}{(Nf_{\text{PIV}}/f)} \sum_{i=1}^{Nf_{\text{PIV}}/f} \left[ \iint u_{\text{inst}}^i(x,y,t) \, dx \, dy \right] \tag{2.14}$$

and

$$\overline{p_y} = \frac{\rho}{(Nf_{PIV}/f)} \sum_{i=1}^{Nf_{PIV}/f} \left[ \iint v_{inst}^i(x,y,t) \ dx \ dy \right]. \tag{2.15}$$

For the hovering case, flow should be in the vertical direction. However, hovering Antarctic krill generally achieve this by orienting their bodies 20°–30° above horizontal [8]. The magnitude and orientation angle ($\delta$) of cycle-averaged momentum per unit width were calculated using the below equations

$$\bar{p} = \sqrt{\overline{p_x}^2 + \overline{p_y}^2} \tag{2.16}$$

and

$$\delta = \tan^{-1}\left(\frac{\overline{p_y}}{\overline{p_x}}\right). \tag{2.17}$$

Error bars representing ±1 s.d. were used to show cycle-to-cycle variation in plots of time-varying and cycle-averaged total momentum ($p_x$, $p_y$, $\overline{p_x}$, $\overline{p_y}$, $\bar{p}$, $\delta$).

# 3. Results

## 3.1. Flow generated by two paddles

Flow generated during PS by synchronous, periodic motion of two paddles consists of co-rotating vortices near the tip of each paddle (figure 3a–d). During the first half of PS, shear layers form near the tip of each paddle (figure 3b), which then roll up into negatively signed (clockwise) vortices (figure 3c). These co-rotating vortices are shed from the paddles near the end of PS (figure 3d), and propagate below the paddles with further progression of the stroke cycle. Viscous dissipation of these vortices occurs subsequently, as evidenced by decreasing vorticity magnitude of the negatively signed vortices in figure 3d–h. Similar to PS, an oppositely signed (counterclockwise) vortex is generated from the tip of each paddle during RS (figure 3f). The interaction of the counterclockwise pair of vortices in RS with the previously shed clockwise vortices from PS (figure 3g) results in generating a jet primarily in the horizontal orientation (figure 3h). Vortex formation and propagation during PS and RS of the right paddle are indicated by dashed red boxes in figure 3.

Metachronal paddling allows for the formation of counter-rotating vortices at the tips of adjacent paddles. The interaction of these counter-rotating vortices formed during metachronal paddling at a phase lag of 16.7% results in the formation of an angled jet moving away from the body (figure 4). The right paddle leads the PS, while the left paddle is phase-delayed in starting its PS. The shear layer formed by PS of the right paddle rolls up into a clockwise vortex that detaches from the tip near the end of PS (figure 4c). However, the left paddle does not start PS until 60% PS of the right paddle (figure 4c). As a result of the phase delay, the clockwise vortex formed at the tip of the right paddle is stronger in magnitude compared with the co-rotating (clockwise) vortex formed at the tip of the left paddle. The shed vortex from the right paddle at the end of its PS (figure 4d) tailors the flow more downward when compared with the same phase point in the synchronous case (figure 3d). RS of the right paddle at 16.7% phase lag generates a shear layer with oppositely signed vorticity compared with that of the left paddle, which completes its PS during the RS of the right paddle (figure 4e,f). The interaction of the counter-rotating vortices shed from both paddle tips occurs at 60% RS of the right paddle (figure 4g), generating a bulk flow that moves downward in the head-to-tail direction (figure 4h). Vortex formation and propagation during PS and RS of the right paddle are indicated by dashed red boxes in figure 4.

Phase lag influences the time-varying geometry of each paddle, which causes the jet to orient in different directions. As phase lag increases from 16.7 to 33.3%, the paddles come in proximity of each other in positions closer to the vertical, generating a more vertical jet at higher $\phi$ (figures 4, 5 and 6). The vertical positions of the counter-rotating vortices formed during PS of the leading (right) paddle also change with phase lag, such that their cores are nearly at the same vertical position at 33.3% (figure 6a–d) when compared with 25% (figure 5a–d). The clockwise vortex shed from the end of PS of the leading paddle at 25% phase lag (figure 5d) dissipates before the end of its RS (figure 5h). By contrast, the clockwise vortex shed from the leading paddle at 33.3% phase lag interacts with the co-signed vortex formed at the tip of the trailing (left) paddle (figure 6e,f). The PS of the trailing

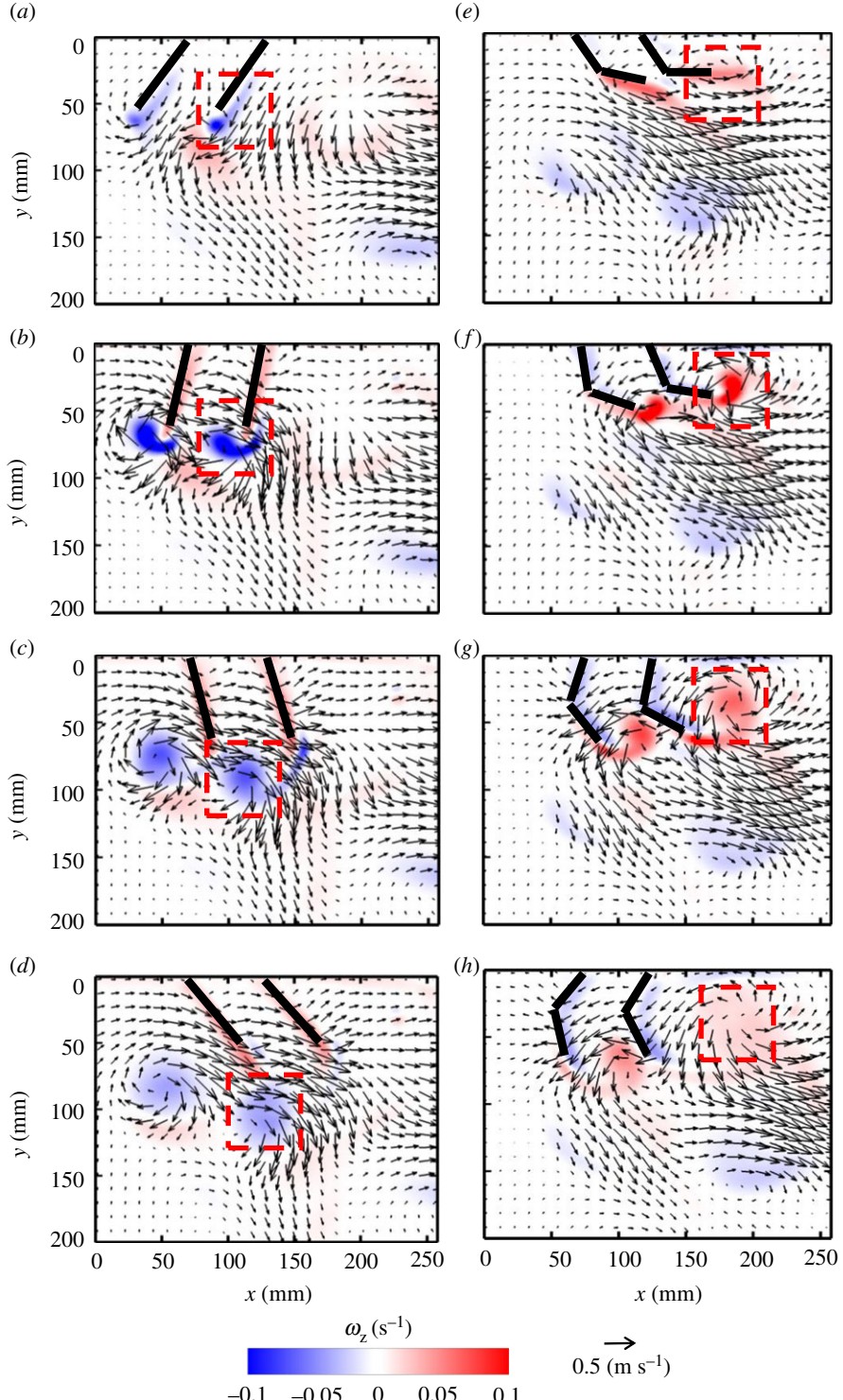

**Figure 3.** Velocity vector fields overlaid on out-of-plane z-vorticity ($\omega_z$) contours for synchronous paddling (0% phase lag) of two paddles at $Re = 250$: (a) 20% PS, (b) 40% PS, (c) 60% PS, (d) 80% PS, (e) 20% RS, (f) 40% RS, (g) 60% RS and (h) 80% RS. Red colour represents counterclockwise vorticity, while blue represents clockwise vorticity. %PS and %RS are referenced with respect to the right-most paddle that is near the tail end of the model. $Re$ was calculated using equation (2.4) and z-vorticity ($\omega_z$) was calculated using equation (2.7). Red boxes indicate the vortices formed during PS (a–d) and RS (e–h) of the right paddle.

paddle starts later with increasing phase lag, roughly at 60% PS (figure 4c), 80% PS (figure 5d) and 20% RS (figure 6e) of the leading paddle. This delay in start of the PS of the trailing paddle allows the shed clockwise vortex following PS of the leading paddle to increase its vorticity by interaction with the co-signed vortex formed at the tip of the trailing paddle (compare figures 4e,f, 5e,f and 6e,f).

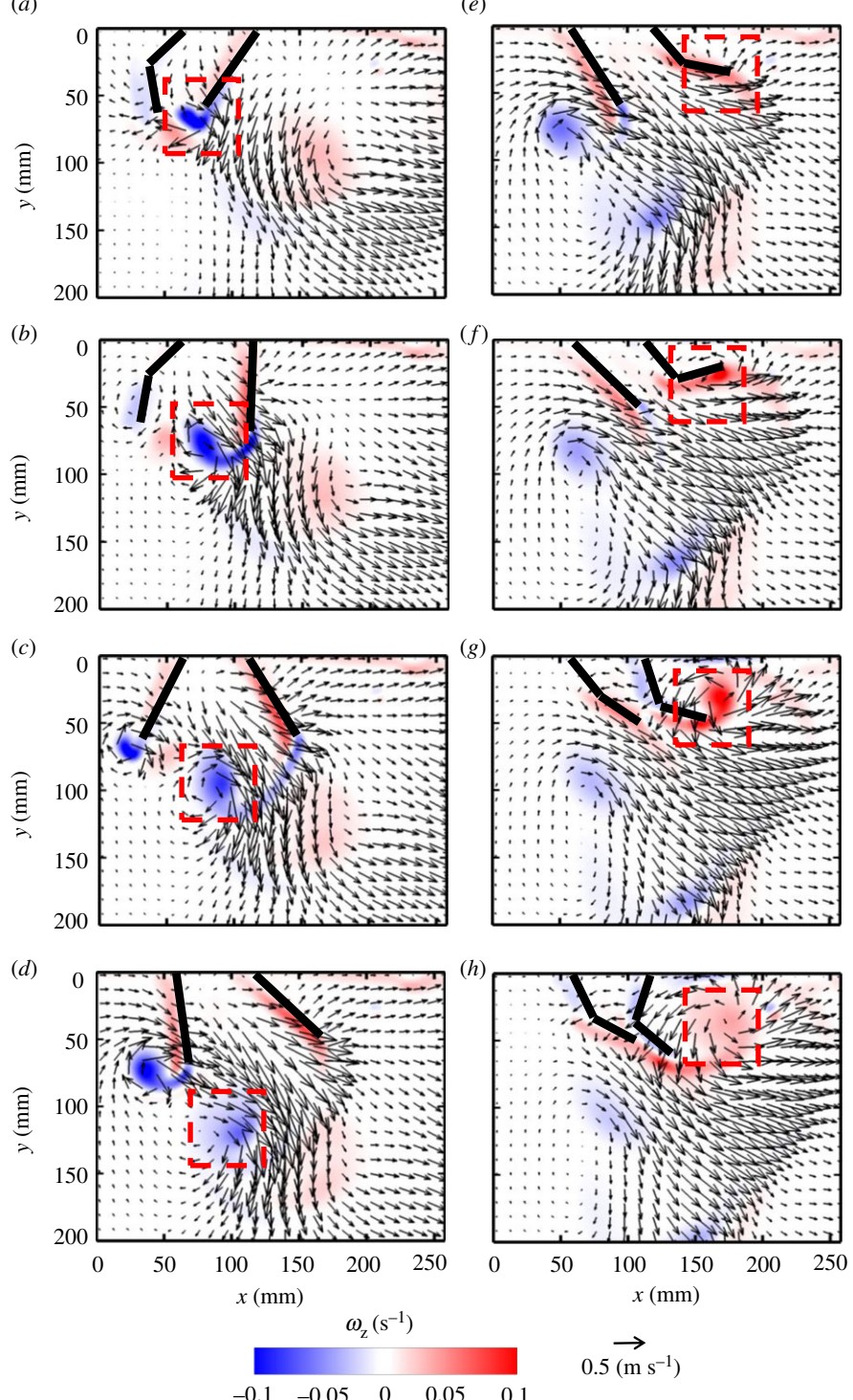

**Figure 4.** Velocity vector fields overlaid on out-of-plane $z$-vorticity ($\omega_z$) contours for metachronal paddling of two paddles at 16.7% phase lag, at $Re = 250$. Definitions of contour colouring, % PS and % RS are the same as in figure 3. Red boxes indicate the vortices formed during PS ($a$–$d$) and RS ($e$–$h$) of the right paddle.

## 3.2. Flow generated by four paddles

With four paddles in synchronous motion at $Re = 250$, four co-rotating vortices are formed in early PS (figure 7$b$) and mid-RS (figure 7$g$). The signs of the vortices formed in PS are opposite to those formed in RS, due to the reversal in rotational direction. The vortex formed by the left-most paddle (P1 in figure 2$a$) in PS is shed at the end of PS (figure 7$d$). The cumulative effect of all PS vortices

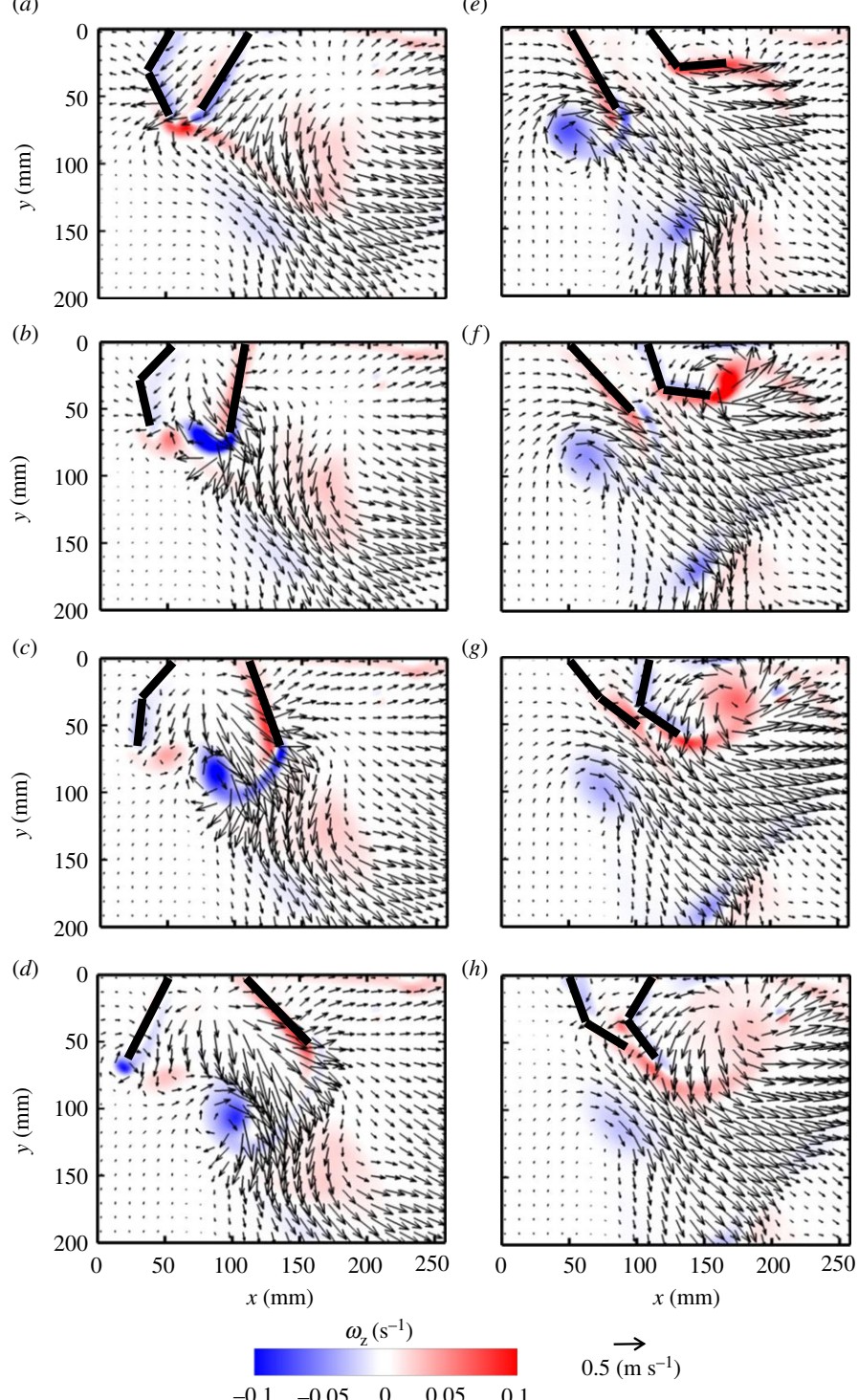

**Figure 5.** Velocity vector fields overlaid on out-of-plane $z$-vorticity ($\omega_z$) contours for metachronal paddling of two paddles at 25% phase lag, at $Re = 250$. Definitions of contour colouring, % PS and % RS are the same as in figure 3.

results in directing the flow downward as well as horizontally to the right. Bending of the paddles early in RS (figure 7$e,f$) does not obstruct the left to right flow generated in PS. The rowing motion in RS forces fluid vertically away from the body, while the hinged nature of the paddles allows flow to propagate towards the tail of the model.

For $Re = 250$ at 25% phase lag (figure 8), a series of small-scale jets form as each pair of paddles come near each other, merging to sustain a continuous large-scale vertical jet (indicated by a dashed red box in figure 8$a$). At several points in the cycle, successive pairs of adjacent paddles move away from each other,

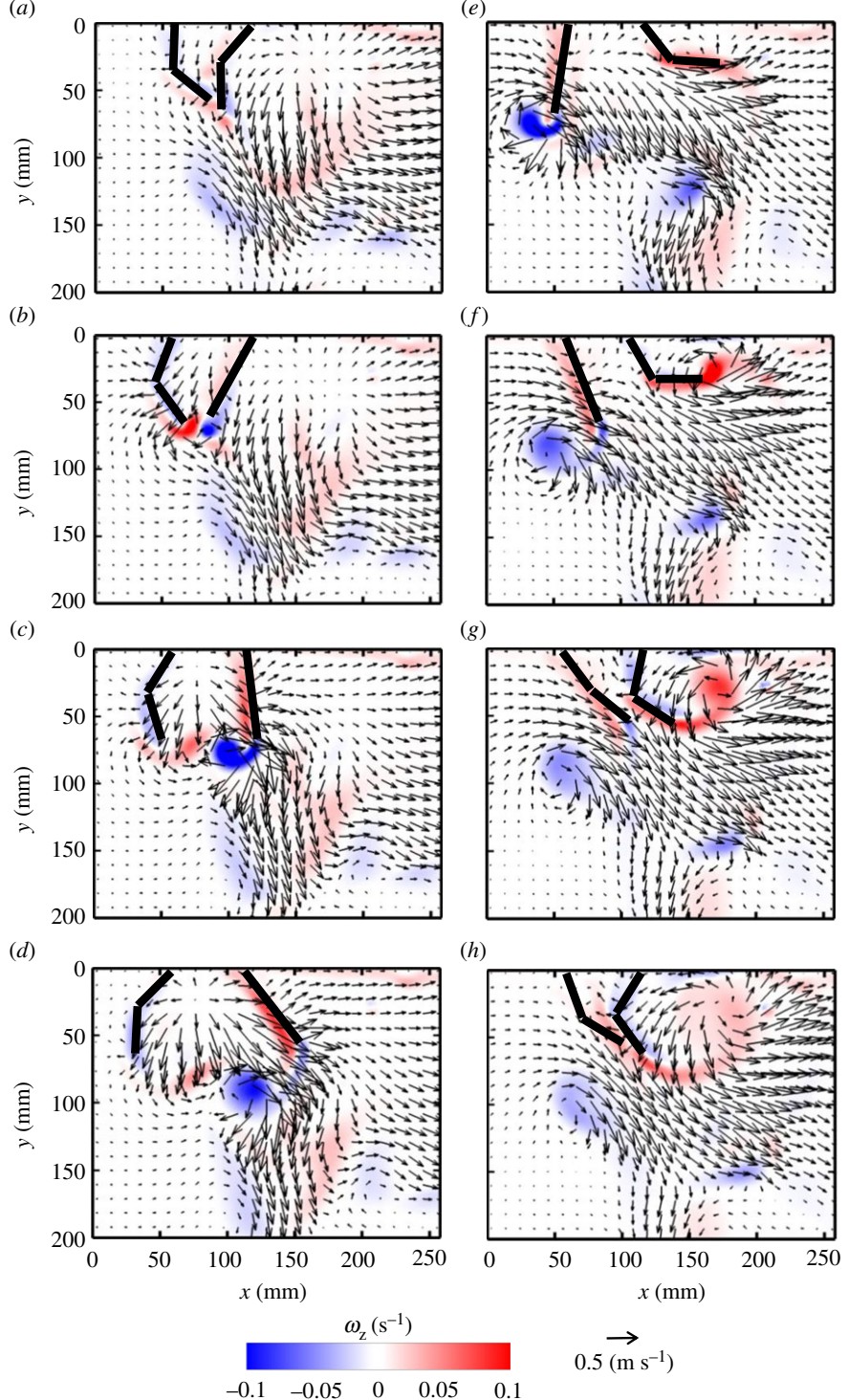

**Figure 6.** Velocity vector fields overlaid on out-of-plane z-vorticity ($\omega_z$) contours for metachronal paddling of two paddles at 33.3% phase lag, at $Re = 250$. Definitions of contour colouring, % PS and % RS are the same as in figure 3.

generating counter-rotating vortex pairs (e.g. P3–P4 in figure 8b; P2–P3 in figure 8e; P1–P2 in figure 8g). Pairs of adjacent paddles also move in the same direction at points across the stroke cycle, generating vortices at their tips (e.g. P2–P3 in figure 8c; P1–P2 in figure 8e; P3–P4 in figure 8h). The proximity of the tips of the paddles moving in the same direction does not allow the production of vortex pairs, and a single vortex develops (e.g. from P2 and P3, both in RS, in figure 8c). Finally, pairs of adjacent paddles move towards each other at different points in the stroke cycle (e.g. P2–P3 in figure 8h; P1–P2 in figure 8b; P3–P4 in figure 8g). Similar to the case of two paddles, a counter-rotating vortex pair

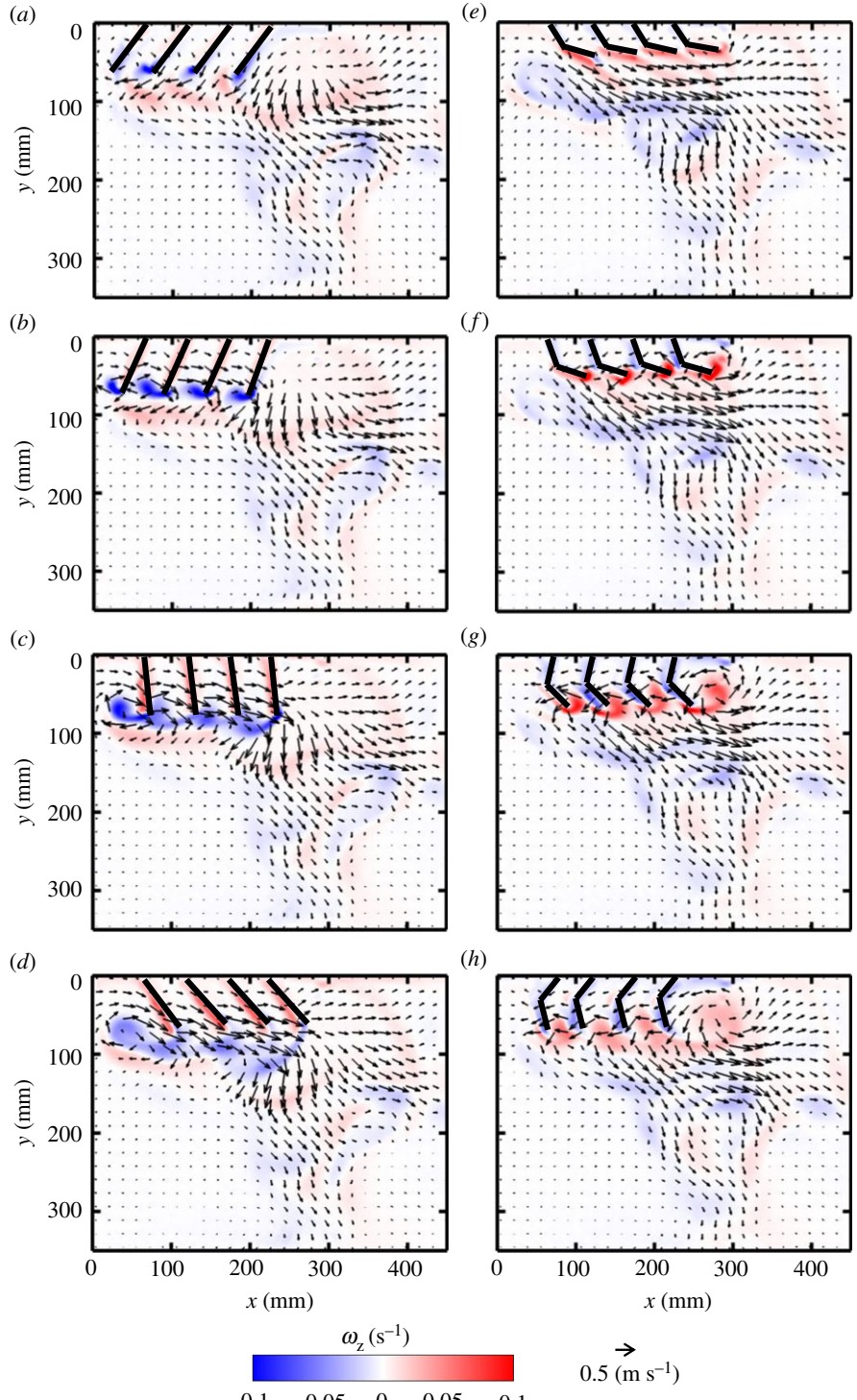

**Figure 7.** Velocity vector fields overlaid on out-of-plane $z$-vorticity ($\omega_z$) contours for synchronous paddling (0% phase lag) of four paddles at $Re = 250$: (*a*) 20% PS, (*b*) 40% PS, (*c*) 60% PS, (*d*) 80% PS, (*e*) 20% RS, (*f*) 40% RS, (*g*) 60% RS and (*h*) 80% RS. Red colouring represents counterclockwise vorticity, while blue represents clockwise vorticity. % PS and % RS are referenced with respect to the right-most paddle (P4 in the inset of figure 2*a*) that is near the tail-end of the model.

forms at the tips of paddles moving towards each other. However, one of these vortices merges with the co-signed vortex formed by the neighbouring paddle moving in the same direction (e.g. in figure 8*b*, the counterclockwise vortex from P2 merges with vortex from P3).

Clockwise vortices are formed with increasing stroke cycle time, starting with the right-most paddle P4 (figure 8*b*), followed by P3 (figure 8*e*), P2 (figure 8*g*) and P1 (figure 8*a*). Vortices shed from the tips of

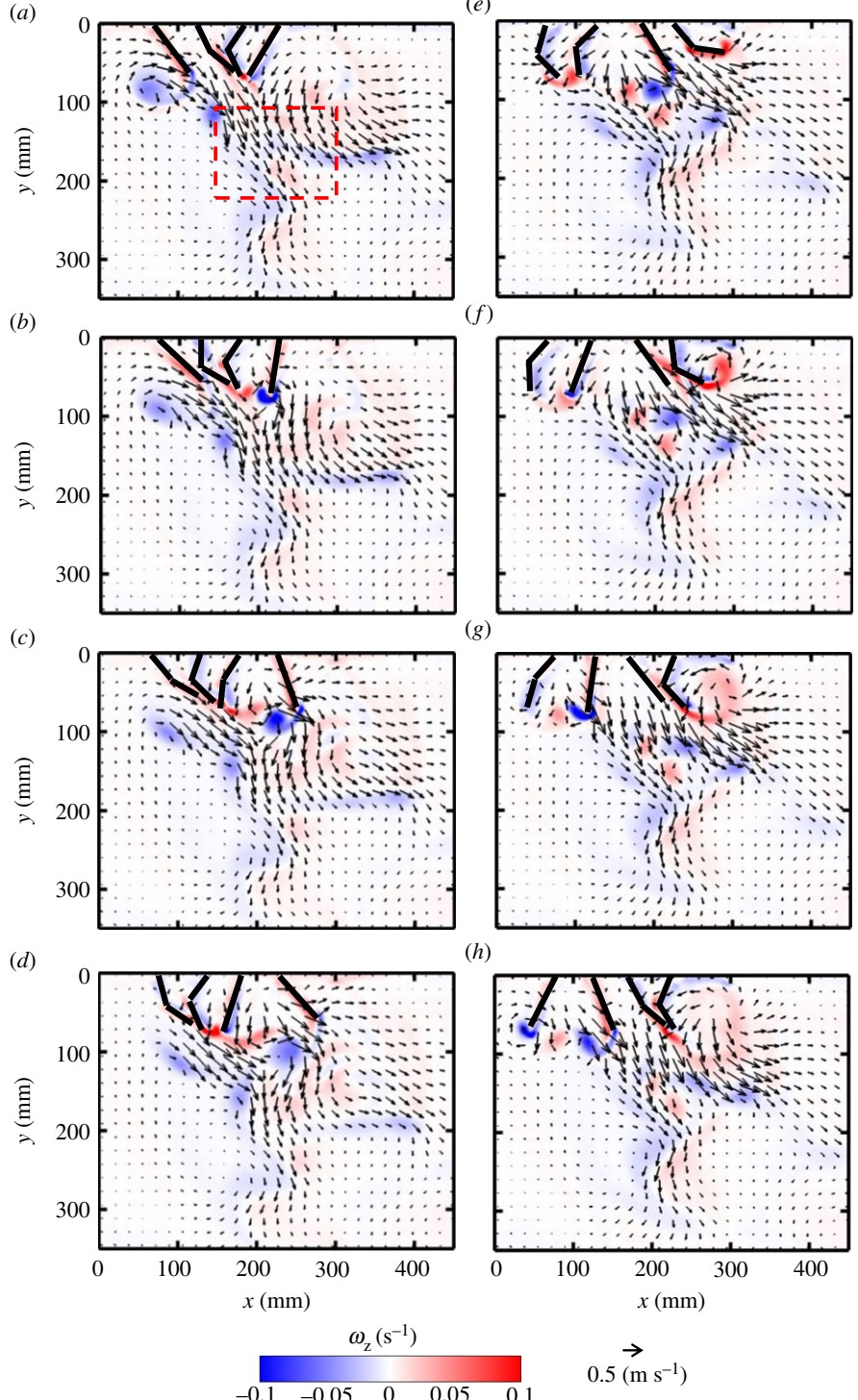

**Figure 8.** Velocity vector fields overlaid on out-of-plane $z$-vorticity ($\omega_z$) contours for metachronal paddling of four paddles at 25% phase lag, at $Re = 250$: (a) 20% PS, (b) 40% PS, (c) 60% PS, (d) 80% PS, (e) 20% RS, (f) 40% RS, (g) 60% RS and (h) 80% RS. Definitions of contour colouring, % PS and % RS are the same as in figure 7. The red box indicates the large-scale jet sustained by the periodic oscillation of the paddles.

different paddle pairs throughout the cycle move downward and interact, generating a continuous shear flow in the form of a downward jet that is approximately below P3–P4 in PS and below P2–P3 in RS.

Flow fields generated by other phase lags at $Re = 250$ are provided as electronic supplementary material, figure S4). Decreasing phase lag from 25 to 16.7% brings paddles closer and reduces the number of counter-rotating vortex pairs formed from adjacent paddles, particularly by P2 and P3 (compare electronic supplementary material, figure S4A–H and figure 8a–h). A downward jet is

observed during the entire cycle at 16.7% phase lag. Compared with 25%, more horizontal flow is observed during PS of 16.7% case in the head-to-tail direction (electronic supplementary material, figure S4A–D), but more reverse flow is observed during RS (electronic supplementary material, figure S4E–H). Increasing phase lag to 33.3% (electronic supplementary material, figure S4I–P) allows for wider gaps between adjacent paddles moving away from each other. This promotes the formation of counter-rotating vortex pairs in PS (electronic supplementary material, figure S4K) and RS (electronic supplementary material, figure S4N). In contrast with the downward jet below P3–P4 at 25% phase lag (figure 8), the jet generated at 33.3% is wider and covers most of the region below P2–P3–P4 near the body, but tapers away from the body.

The divergence of the 2D velocity fields, in the form of point sources (vectors pointing away from each other) or point sinks (vectors pointing towards each other), can be seen in some instances in the flow fields (figures 3–8, 9 and 10). Flow sources and sinks indicate fluid motion was out of the plane where PIV recordings were obtained. The extent of 3D of the flow was characterized by computing 2D divergence of the velocity field, defined in the below equation (3.1)

$$\nabla \cdot \boldsymbol{U} = \frac{\partial u}{\partial \mathrm{x}} + \frac{\partial v}{\partial \mathrm{y}}. \tag{3.1}$$

A region with positive divergence appears in the velocity field as a point source, while a region with negative divergence appears as a point sink. Divergence was calculated for the velocity fields shown in figures 7–10 (0% phase lag at $Re = 250$ and 25% phase lag at $Re = 250$, 50 and 800, respectively), and are provided as electronic supplementary material, figures S5–S8. Regions of positive divergence were formed behind the tail-most paddle at the end of synchronous recovery (electronic supplementary material, figure S5), as well as between paddles moving away from each other during metachronal motion (electronic supplementary material, figure S6). Regardless of $Re$, positive divergence regions were generated in the same regions for each case, but were of much smaller magnitude at $Re = 50$ (electronic supplementary material, figure S7) and higher magnitude at $Re = 800$ (electronic supplementary material, figure S8), when compared with the divergence at $Re = 250$ (electronic supplementary material, figure S6). Though our paddles were idealized as 2D flat plates, the mechanical system used to drive paddle motion required gaps between the edges of the paddles and the sides of the tank (along the z-axis). These gaps probably allowed for fluid to be entrained from out of the rotational plane of the paddles, resulting in some 3D of the flow. This is supported by a previous study by Kim & Gharib [25], where rotational motion of a single flat plate paddle was shown to result in the generation of a 3D tip vortex that entrains flow from outside of the rotational plane (along the axis of rotation).

## 3.3. Effects of varying Re

With changing $Re$ at a constant phase lag of 25% (figures 8–10), the role of viscous dissipation on metachronal paddling becomes clear. At $Re = 50$, individual jets in the series do not have an opportunity to interact with each other, as they dissipate quickly due to viscosity (figure 9). This markedly limits the vertical propagation of the downward jet compared with higher $Re$. At $Re = 250$, the jets from the three pairs of paddles are able to interact, directing some momentum downward. However, at $Re = 800$, the unsteadiness of the flow is not as present in the far-field, and 300 mm below the base, a near-steady jet is observed in the flow field (figure 10). For other phase lags (0, 16.7, 33.3%), similar viscous dissipation effects were observed with changing $Re$ (see electronic supplementary material, figure S9–S14).

Components of time-varying total linear momentum in the flow field were calculated for each $Re$ and phase lag (figure 11). The horizontal component of total momentum ($p_x$) is unsteady throughout the cycle (figure 11d–f), while the vertical component of momentum ($p_y$) remains nearly constant (figure 11g–i). Both $p_x$ and $p_y$ were higher at $Re = 250$ and $Re = 800$ than at $Re = 50$, except for $p_y$ in the case of 0% phase offset. However, momentum generated by the $Re = 250$ case (figure 11e,h) were nearly equal to those at $Re = 800$ (figure 11f,i). At $Re = 50$, $p_y$ was found to be monotonically increasing with increasing phase lag, which was not the case for the other $Re$. At $Re = 250$ and $Re = 800$, approximately the range of pleopod-based $Re$ in euphausiids, $p_y$ is comparable for metachronal paddling regardless of phase lag (between 16.7 and 33%), but much lower for synchronous paddling. However, $p_x$ is greater for phase lags between 16.7 and 25%, close to the phase lags reported for

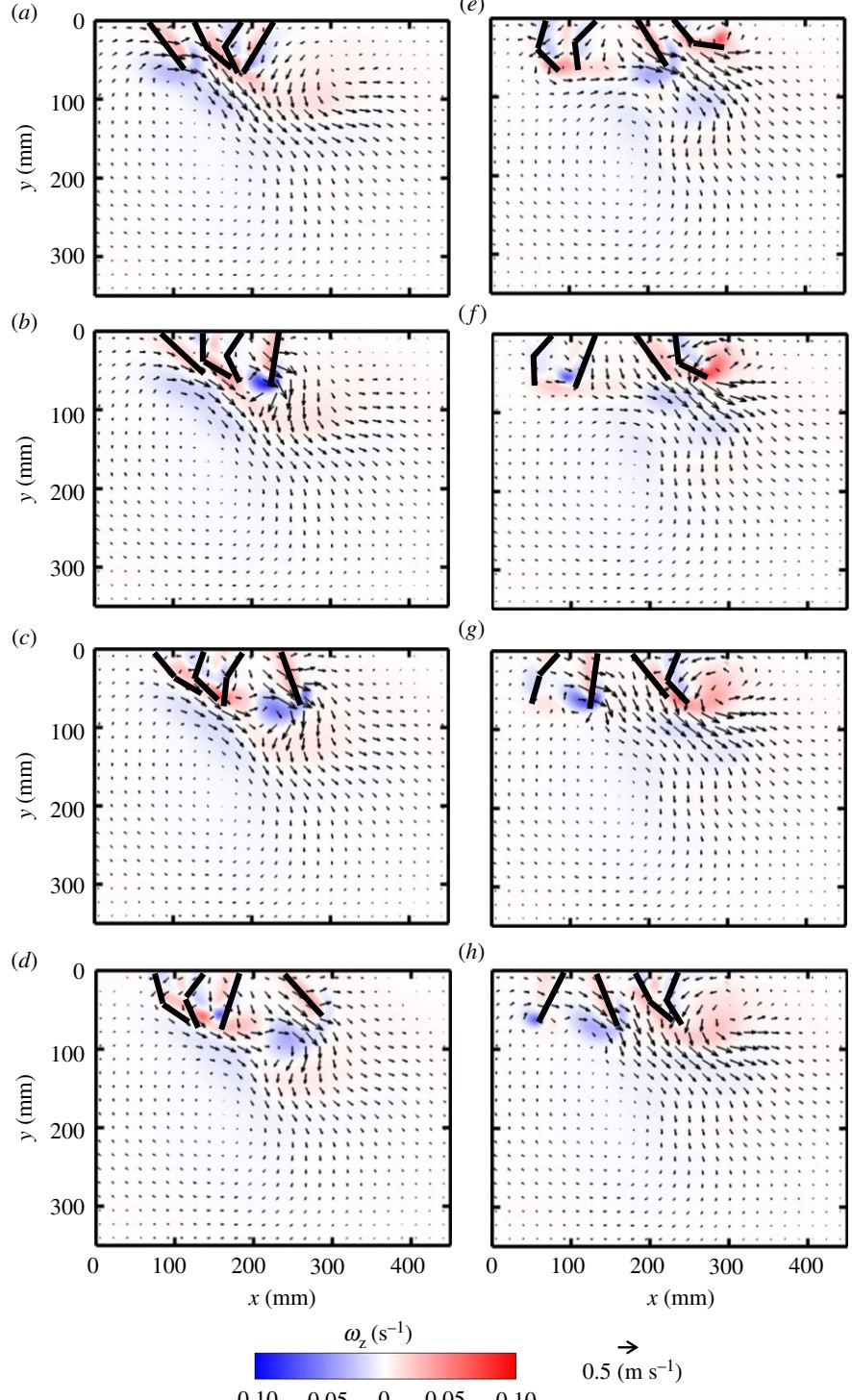

**Figure 9.** Velocity vector fields overlaid on out-of-plane $z$-vorticity ($\omega_z$) contours for metachronal paddling of four paddles at 25% phase lag, at $Re = 50$: (a) 20% PS, (b) 40% PS, (c) 60% PS, (d) 80% PS, (e) 20% RS, (f) 40% RS, (g) 60% RS, (h) 80% RS. Definitions of contour colouring, % PS and % RS are the same as in figure 7.

*E. superba* [8]. In every case, standard deviations were much smaller than the marker sizes, so error bars are not displayed in figure 11.

Momentum fluxes were used to examine the location of force generated in the flow (figure 12). In each case, the highest values of HMF occurred behind the model, while the highest values of VMF occurred slightly below the tips of the fully extended paddles. With increasing phase lag, HMF decreased, whereas VMF increased and remained high farther below the base of the model. Peak flux

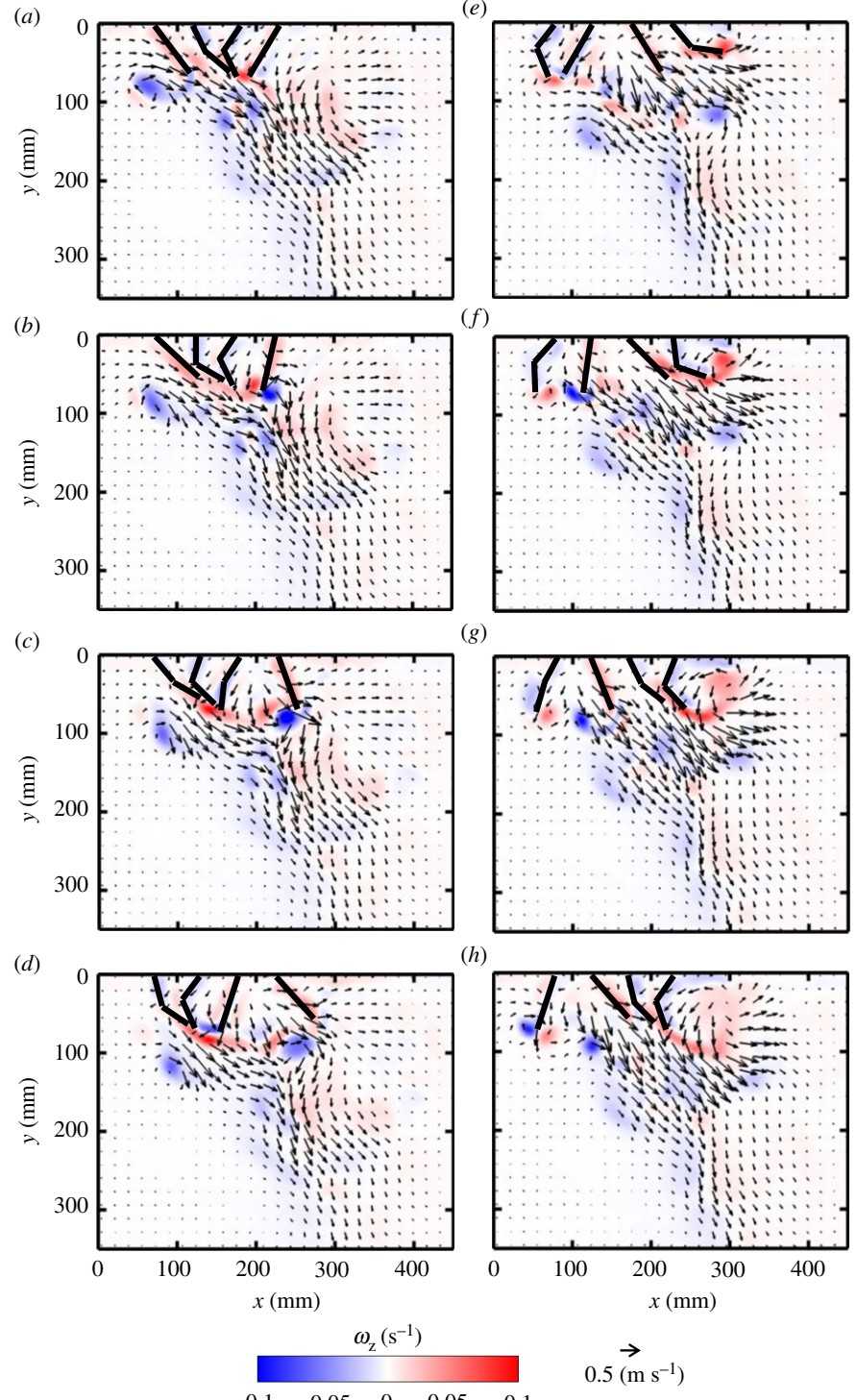

**Figure 10.** Velocity vector fields overlaid on out-of-plane z-vorticity ($\omega_z$) contours for metachronal paddling of four paddles at 25% phase lag, at $Re = 800$: (a) 20% PS, (b) 40% PS, (c) 60% PS, (d) 80% PS, (e) 20% RS, (f) 40% RS, (g) 60% RS and (h) 80% RS. Definitions of contour colouring, % PS and % RS are the same as in figure 7.

values were lowest for $Re = 50$ across all phase lags. $Re = 800$ had highest peak flux values at 16.7 and 25% phase lags, while $Re = 250$ had highest peak flux values at 0 and 33.3% phase lags.

Cycle-averaged total momentum values were used to examine what conditions contribute most to vertical and horizontal flow (figure 13). $Re = 50$ has the lowest values of $\overline{p_x}$, $\overline{p_y}$ and $\bar{p}$ for all phase lags (except for $\overline{p_y}$ at 0% phase lag, as noted previously). For phase lags of 25% or below, no statistical difference was found between momentum generated for $Re = 250$ and $Re = 800$. However,

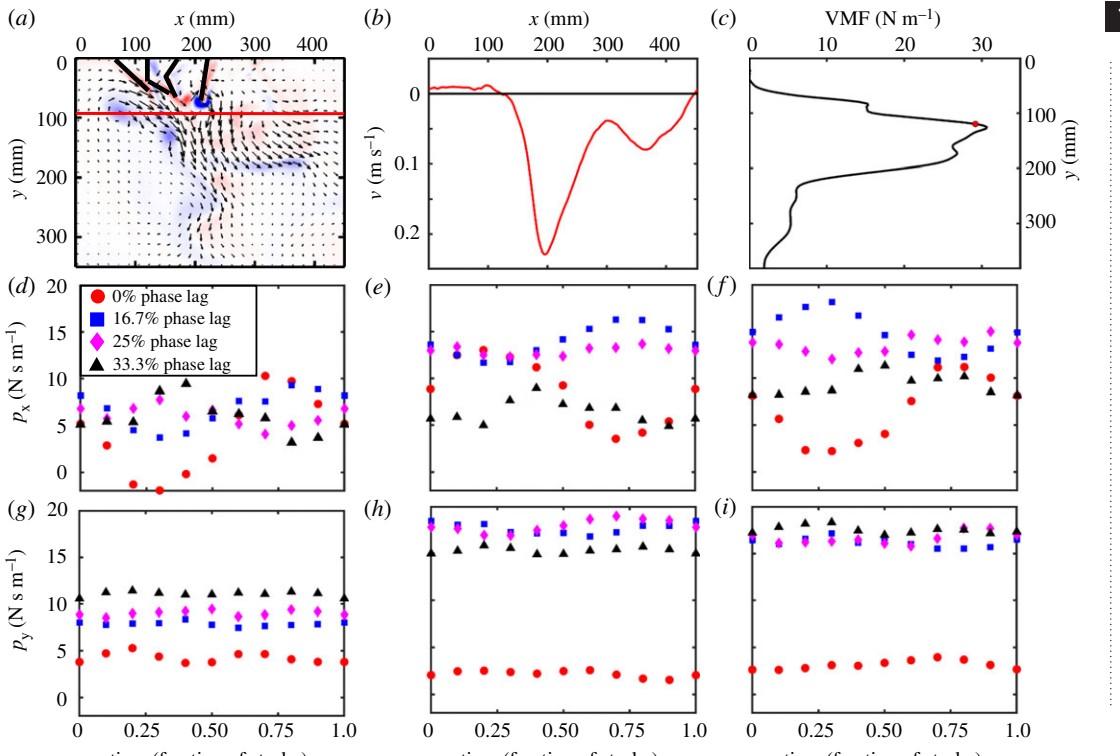

**Figure 11.** Representation of the procedure used to calculate cycle-averaged momentum flux per unit width of the paddle in (*c*), and horizontal and vertical components of phase-averaged total momentum in PIV FOV in (*d*–*i*). (*a*) Velocity vector field overlaid on out-of-plane *z*-vorticity ($\omega_z$) contours for $Re = 250$, 25% phase lag, at 40% PS of the right-most paddle (P4 in the inset of figure 2*a*). (*b*) Vertical velocity component (*v*) extracted along the solid red line shown in (*a*). (*c*) VMF calculated using equation (2.13) at horizontal lines drawn for all *y*-locations in the PIV FOV. VMF corresponding to the *y*-location selected in (*a*) is represented by the red circle in (*c*). (*d*–*f*) Horizontal component of total momentum ($p_x$) calculated using equation (2.10) is shown as a function of time and phase lag for: (*d*) $Re = 50$, (*e*) $Re = 250$ and (*f*) $Re = 800$. (*g*–*i*) Vertical component of total momentum per unit width of the paddle ($p_y$), calculated using equation (2.11), is shown as a function of time and phase lag for: (*g*) $Re = 50$, (*h*) $Re = 250$ and (*i*) $Re = 800$. (*a*)–(*c*) were obtained from cycle-averaging velocity vector field data across 30 consecutive cycles. (*d*)–(*i*) were obtained from velocity vector field data that were phase-averaged across 30 consecutive cycles. In each case, standard deviations (represented using error bars) are smaller than the marker size.

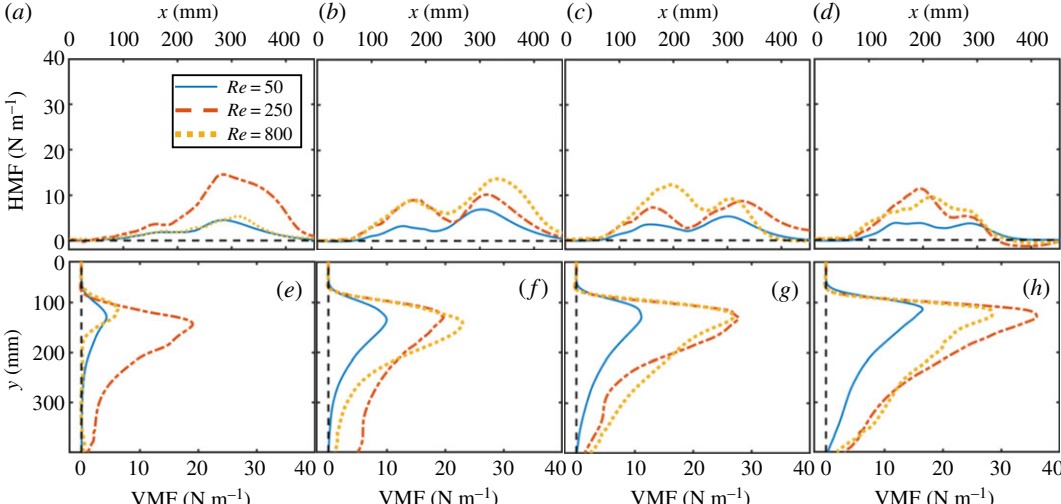

**Figure 12.** Cycle-averaged momentum fluxes as a function of phase lag and varying locations in the flow field. (*a*–*d*) HMF versus *x*-location for (*a*) 0% phase lag, (*b*) 16.7% phase lag, (*c*) 25% phase lag and (*d*) 33.3% phase lag. (*e*–*h*) VMF versus *y*-position for (*e*) 0% phase lag, (*f*) 16.7% phase lag, (*g*) 25% phase lag and (*h*) 33.3% phase lag. HMF was calculated using equation (2.12) and VMF was calculated using equation (2.13). Blue lines represent $Re = 50$, red lines represent $Re = 250$ and yellow lines represent $Re = 800$.

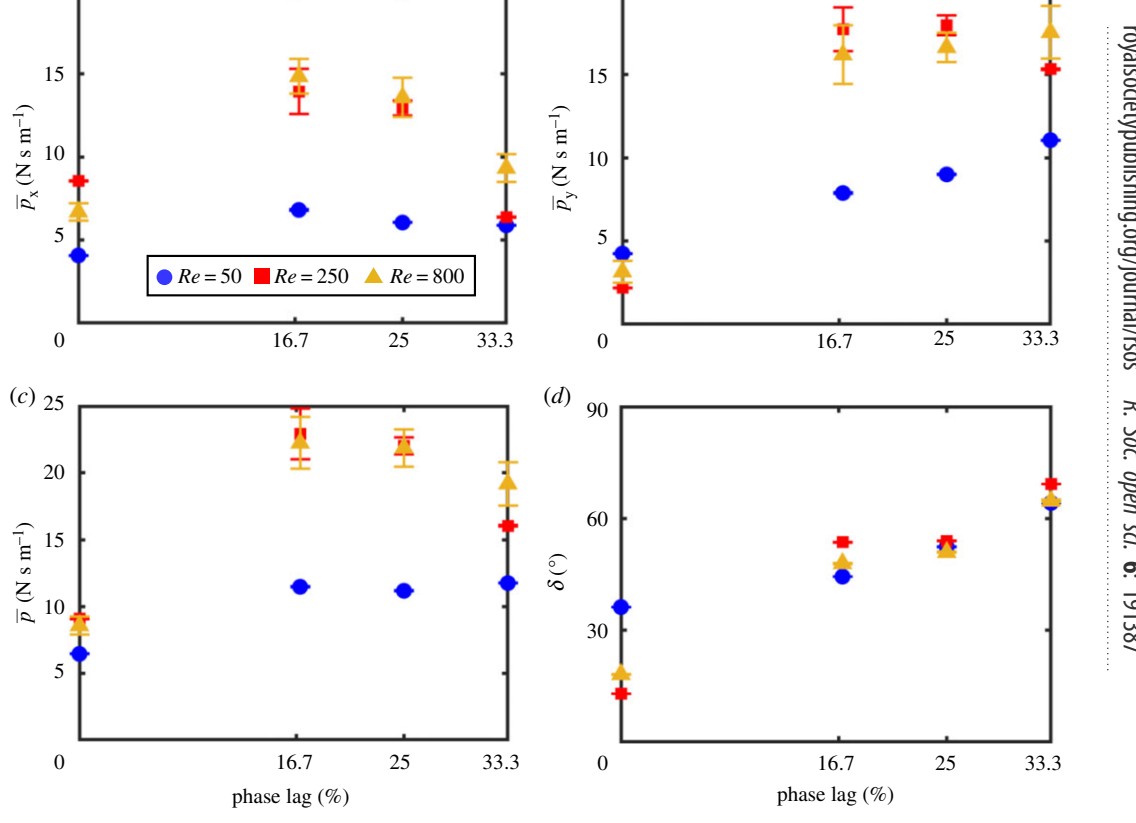

**Figure 13.** Cycle-averaged total momentum characteristics. (a) Horizontal component of cycle-averaged total momentum ($\overline{p_x}$) calculated using equation (2.14). (b) Vertical component of cycle-averaged total momentum ($\overline{p_y}$) calculated using equation (2.15). (c) Magnitude of cycle-averaged total momentum vector ($\overline{p}$), calculated using equation (2.16). (d) Orientation angle ($\delta$) of the cycle-averaged total momentum vector, calculated using equation (2.17).

higher values of $\overline{p_x}$, $\overline{p_y}$ and $\overline{p}$ were observed for 33.3% phase lag at $Re = 800$. The orientation of the momentum vector ($\delta$) changes from mostly horizontal at 0% phase lag to mostly vertical with increasing phase lag for each $Re$ (figure 13d). At 0% phase lag, $Re = 50$ had the most vertical jet, since $Re = 50$ had the highest $\overline{p_y}$ and the lowest $\overline{p}$.

## 4. Discussion

Pelagic crustaceans such as krill have to contend with negative buoyancy due to large tissue density [11]. Kils [11] reported that non-swimming krill can sink by as much as 500 m in 3 h. These animals must therefore maintain their position in the water column through a constant downward transfer of momentum to the surrounding fluid [9,11,26,27]. Previous studies of metachronal swimming using live crustaceans [8–20,28], theoretical models [7,23,28] and numerical simulations [22] have not clarified how oscillation of closely spaced pleopods in the horizontal plane can generate downward momentum [9,21]. From experiments on a tethered robotic platform fitted with flat plate paddles, we found that: (i) metachronal paddling with non-zero phase lag creates adjacent paddle geometries that promote counter-rotating vortex generation; (ii) counter-rotating vortices formed in metachronal paddling interact to generate downward (lift-generating) jets; (iii) synchronous paddling does not promote counter-rotating vortex formation within either half-stroke (PS or RS), resulting in primarily horizontal flow generation; and (iv) increasing $Re$ by lowering viscosity increases the extent of downward flow on account of diminished viscous dissipation. While the angular orientation of the large-scale downward jet depends on the phase lag, asymmetric bending of hinged paddles in RS directs the jet towards the tail end of the model. Metachronal paddling is thus capable of producing both lift (vertical) and thrust (horizontal) forces needed for fast forward swimming and hovering gaits observed in freely swimming krill [8].

Despite the idealization of 3D crustacean pleopods as 2D flat plate paddles, our results show the development of angled downward jets that have been observed in hovering krill [9]. Phase lag acted as the primary driver in transferring force imparted by paddle oscillation in the horizontal plane to the downward direction. Zhang *et al.* [22] proposed that asymmetry in geometries of neighbouring limbs, established by non-zero phase lag, can allow for more fluid volume to be captured in PS as opposed to RS. We observed total momentum was generally larger in non-zero phase lag conditions when compared with synchronous paddling (figure 11*d*–*i*). Time-varying geometries of adjacent paddles that were either moving away from or towards each other promoted the formation of counter-rotating tip vortices, similar to the clap and fling mechanism used in flapping flight of tiny insects at low *Re* [29]. By contrast, counter-rotating vortices were generated in synchronous paddling only when overall oscillation direction changed (i.e. between PS and RS). The interaction of a counter-rotating vortex pair that was formed by paddles moving towards each other directed flow downward from the body. The vortex–vortex interaction observed in paddles moving away from each other directed some fluid towards the gaps (e.g. P3–P4 in figure 9*d*). However, unlike clap and fling where a paddle would move entirely out-of-phase to the neighbouring paddle (50% phase lag), the relatively smaller phase lags reduced the extent to which the jet was directed downward (contributing to downward momentum).

Increasing phase lag between pairs of paddles resulted in increasing the angular orientation of the momentum jet (more vertical). This was due to the increase in the gaps between neighbouring paddles that promoted more symmetry of the counter-rotating vortex pairs. Though angular orientation increased, the total momentum of the jet ($\bar{p}$) showed a more complex dependence on phase lag. Phase lags of 16.7 and 25% produced the highest $\bar{p}$, and further increasing phase lag to 33.3% resulted in decreasing $\bar{p}$. Dissipative interactions between the larger counter-rotating vortices generated at 33.3% could probably be the reason for the drop in momentum. Interestingly, hovering *E. superba* operate with phase lags between 16.7 and 25% [8], corresponding to the range that generated the most momentum in our model. *Euphausia superba* reorient their body angle during hovering to be in the range of 25–50° [8]. Considering the angular orientation ($\delta$) of the total momentum vector generated in our model (phase lags of 16.7% and 25%) ranged between 50 to 60°, reorienting the body by 25 to 50° would result in almost vertical jets, which would be helpful in maintaining the position of a krill in the water column.

The Antarctic krill stores low-density lipids to increase buoyancy, and the extent of this storage varies seasonally [9]. Further, body mass changes with development stage, so that larval krill would be more positively buoyant than adults. Finally, dynamic viscosity of water increases with decreasing temperature. Change in *Re* can thus be expected between different developmental stages (changing pleopod length and swimming speed) and between different seasons (changing temperature). To the best of our knowledge, previous studies of metachronal swimming have not examined how change in *Re* impacts the generation of downward momentum. We found that increasing *Re* from 50 to 250 resulted in generating larger vertical momentum than horizontal momentum. The increased viscosity at lower *Re* dissipates vortices and prevents downward propagation of the jet. For supporting weight and maintaining position in the water, larval krill (lower *Re*) would probably not need to generate as much vertical momentum as adults (higher *Re*) due to their relative buoyancy difference.

Crustacean pleopods have hinged articulations that allow the lower portion of the pleopods (e.g. endopodites in krill pleopods) to fold inward during RS, and expand outward during PS [8]. Zhang *et al.* [22] modelled PS–RS asymmetry by introducing permeability in their paddles during RS. However, they did not observe large-scale angled downward jets as in our study. The hinges positioned mid-length of paddles in our model permitted differential bending in RS. This introduced asymmetry in flow generated during PS and RS, and directed bulk flow towards the tail of the model. Bending of paddles about the hinges in RS helped to tailor small-scale jets between pairs of paddles towards the tail, which coalesce to form an angled downward jet. The inclusion of hinges in the paddles created flow asymmetry even in synchronous paddling by permitting flow generated during PS to move horizontally towards the tail during RS.

Our model was designed with an inter-paddle gap to paddle length ratio (inter-paddle distance to paddle length) similar to Pacific krill [8]. Murphy *et al.* [8] observed a narrow range of gap to length ratio among metachronal swimmers, and argued that this ratio could affect the hydrodynamic interactions that occur between neighbouring appendages. Our observations show that the close spacing of adjacent paddles promoted the interaction of counter-rotating tip vortices and led to the development of large-scale downward jets. In terms of stroke kinematics, freely swimming crustaceans can independently vary the SA and phase lag of each pleopod pair [8]. Our model was limited to maintaining a constant SA and the same phase lag between each pair of paddles. Finally, we were not

able to assess swimming performance in the tethered model. Further studies are needed to examine swimming performance across varying phase lag and *Re*, and understand the hydrodynamic role of gap to length ratio under non-uniform stroke kinematics (phase lag and SA) of pleopod pairs.

# 5. Conclusion

Metachronal rowing of multiple, closely spaced flat plate paddles about the horizontal plane at biologically intermediate *Re* ranging from 50 to 800 generated counter-signed vortices during the entire stroke cycle. By contrast, synchronous rowing generated co-signed vortices until direction of oscillation was reversed at the end of each half-stroke. Vortex interactions in metachronal paddling resulted in synthesis of small-scale jets between paddles that coalesce to form large-scale, angled downward jets. Differential bending of paddles in recovery, due to inclusion of hinges, tailored the flow in the head-to-tail direction. Decreasing *Re* resulted in diminishing downward propagation of the jets away from the paddles. These findings show that metachronal paddling is capable of generating downward momentum needed for negatively buoyant crustaceans to maintain their position in the water column during hovering and fast forward swimming.

Data accessibility. Data used in figures are available within Figshare: https://doi.org/10.6084/m9.figshare.8298860.v2 [30].
Authors' contributions. M.P.F., H.K.L., M.S. and A.S. conceived of and designed the study. M.P.F. and H.K.L. acquired experimental data. M.P.F. and A.S. analysed the data and drafted the manuscript. A.S., H.K.L. and M.S. critically revised the manuscript. All authors gave final approval for publication and agree to be held accountable for the work performed therein.
Competing interests. We declare we have no competing interests.
Funding. This work was supported by the National Science Foundation (grant nos. CBET 1706762 and CBET 1512071 to A.S.).
Acknowledgements. We thank the referees for their helpful comments.

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
