## [Reviewer comments · Royal Society Open Science]

Review History

RSOS-191387.R0 (Original submission)

Review form: Reviewer 1

Is the manuscript scientifically sound in its present form?

Yes

Are the interpretations and conclusions justified by the results?

Yes

Is the language acceptable?

Yes

Do you have any ethical concerns with this paper?

No

Have you any concerns about statistical analyses in this paper?

No

Recommendation?

Accept as is

Comments to the Author(s)

The effects of varying kinematics of metachronal wave paddling on the resulting flow are nicely illustrated in this manuscript.

I enjoyed reading it and think it an important contribution to our knowledge of this method of locomotion which is so widespread throughout the animal kingdom.

The paper is generally clear and well written. The results of the experiments with the robotic model confirm what experts would have expected and I expect it to enter the textbooks on animal locomotion from now on.

I have only two minor remarks:

The figures showing the flow visualization results can be made much clearer. For example, by decreasing the number of arrows and increasing the size of the figures (figures 3 – 9 and 10A). Hinge kinematics including the limits to the hinge angles should be given in the experimental methods section (2).

Review form: Reviewer 2**Is the manuscript scientifically sound in its present form?**

Yes

Are the interpretations and conclusions justified by the results?

Yes

Is the language acceptable?

Yes

Do you have any ethical concerns with this paper?

No

Have you any concerns about statistical analyses in this paper?

No

Recommendation?

Accept with minor revision (please list in comments)

Comments to the Author(s)

Experiments are reported in which fluid flow is driven around a series of paddles. Motivated by hovering and swimming krill, the authors set up two or four paddles to oscillate with different phase lags and Reynolds numbers. Their main findings are that, with more phase lag the adjacent paddles produce counter-rotating vortices and downward-pointing jets, while the jets diminish at lower Reynolds numbers. I believe the research is original and the conclusions are generally well supported by the experimental data.

My main concern is that, while the experiments are designed to produce 2D flow fields, the plots show regions with arrows converging or diverging to a point (e.g. to the right of the paddles in Fig 3A, 3B, etc.), implying 3D flow. Could the authors quantify this (e.g. by computing the divergence in 2D) and discuss possible reasons why the flow is going in or out of the laser sheet?

In addition, it was not always clear which specific vortices and jets the main text referred to in each of the figure panels. It would be helpful if the key regions or set of arrows could be somehow highlighted or distinguished, e.g. with the use of additional markers or pointers.

Once these issues are resolved I would recommend publication in Royal Society Open Science.

Decision letter (RSOS-191387.R0)

08-Sep-2019

Dear Dr Santhanakrishnan

On behalf of the Editors, I am pleased to inform you that your Manuscript RSOS-191387 entitled "Hydrodynamics of metachronal paddling: effects of varying Reynolds number and phase lag" has been accepted for publication in Royal Society Open Science subject to minor revision in accordance with the referee suggestions. Please find the referees' comments at the end of this email.

The reviewers and handling editors have recommended publication, but also suggest some minor revisions to your manuscript. Therefore, I invite you to respond to the comments and revise your manuscript.

- Ethics statement

- Data accessibility

If you wish to submit your supporting data or code to Dryad (<http://datadryad.org/>), or modify your current submission to dryad, please use the following link:
<http://datadryad.org/submit?journalID=RSOS&manu=RSOS-191387>

- Competing interests

- Authors' contributions

All submissions, other than those with a single author, must include an Authors' Contributions section which individually lists the specific contribution of each author. The list of Authors

should meet all of the following criteria; 1) substantial contributions to conception and design, or acquisition of data, or analysis and interpretation of data; 2) drafting the article or revising it critically for important intellectual content; and 3) final approval of the version to be published.

- Acknowledgements

- Funding statement

Because the schedule for publication is very tight, it is a condition of publication that you submit the revised version of your manuscript before 17-Sep-2019. Please note that the revision deadline will expire at 00.00am on this date. If you do not think you will be able to meet this date please let me know immediately.

- 1) A text file of the manuscript (tex, txt, rtf, docx or doc), references, tables (including captions) and figure captions. Do not upload a PDF as your "Main Document";

- 2) A separate electronic file of each figure (EPS or print-quality PDF preferred (either format should be produced directly from original creation package), or original software format);
- 3) Included a 100 word media summary of your paper when requested at submission. Please ensure you have entered correct contact details (email, institution and telephone) in your user account;
- 4) Included the raw data to support the claims made in your paper. You can either include your data as electronic supplementary material or upload to a repository and include the relevant doi within your manuscript. Make sure it is clear in your data accessibility statement how the data can be accessed;
- 5) All supplementary materials accompanying an accepted article will be treated as in their final form. Note that the Royal Society will neither edit nor typeset supplementary material and it will be hosted as provided. Please ensure that the supplementary material includes the paper details where possible (authors, article title, journal name).

on behalf of Professor Brooke Flammang (Associate Editor) and R. Kerry Rowe (Subject Editor)
openscience@royalsociety.org

Associate Editor Comments to Author (Professor Brooke Flammang):

In general both reviewers had positive comments about the manuscript. The greatest need for improvement is the suggested minor edits to make the figures more clear. To make the analysis complete, please also address the question on divergence of flow outside of the 2D plane of interrogation.

Reviewer comments to Author:

Reviewer: 1

Comments to the Author(s)

The effects of varying kinematics of metachronal wave paddling on the resulting flow are nicely illustrated in this manuscript.

I enjoyed reading it and think it an important contribution to our knowledge of this method of locomotion which is so widespread throughout the animal kingdom.

The paper is generally clear and well written. The results of the experiments with the robotic model confirm what experts would have expected and I expect it to enter the textbooks on animal locomotion from now on.

I have only two minor remarks:

The figures showing the flow visualization results can be made much clearer. For example, by decreasing the number of arrows and increasing the size of the figures (figures 3 – 9 and 10A).

Hinge kinematics including the limits to the hinge angles should be given in the experimental methods section (2).

Reviewer: 2

Comments to the Author(s)

Experiments are reported in which fluid flow is driven around a series of paddles. Motivated by hovering and swimming krill, the authors set up two or four paddles to oscillate with different phase lags and Reynolds numbers. Their main findings are that, with more phase lag the adjacent paddles produce counter-rotating vortices and downward-pointing jets, while the jets diminish at lower Reynolds numbers. I believe the research is original and the conclusions are generally well supported by the experimental data.

My main concern is that, while the experiments are designed to produce 2D flow fields, the plots show regions with arrows converging or diverging to a point (e.g. to the right of the paddles in Fig 3A, 3B, etc.), implying 3D flow. Could the authors quantify this (e.g. by computing the divergence in 2D) and discuss possible reasons why the flow is going in or out of the laser sheet?

In addition, it was not always clear which specific vortices and jets the main text referred to in each of the figure panels. It would be helpful if the key regions or set of arrows could be somehow highlighted or distinguished, e.g. with the use of additional markers or pointers.

Once these issues are resolved I would recommend publication in Royal Society Open Science.

Author's Response to Decision Letter for (RSOS-191387.R0)

See Appendix A.

Decision letter (RSOS-191387.R1)

20-Sep-2019

Dear Dr Santhanakrishnan,

I am pleased to inform you that your manuscript entitled "Hydrodynamics of metachronal paddling: effects of varying Reynolds number and phase lag" is now accepted for publication in Royal Society Open Science.

on behalf of Professor Brooke Flammang (Associate Editor) and R. Kerry Rowe (Subject Editor)
openscience@royalsociety.org

Appendix A

Associate Editor:

In general both reviewers had positive comments about the manuscript. The greatest need for improvement is the suggested minor edits to make the figures more clear. To make the analysis complete, please also address the question on divergence of flow outside of the 2D plane of interrogation

Most of the figures in the manuscript have been redone for clarity.

We have increased figure size and decreased vector density in Figures 3-8, 9, and 10A in order to improve clarity of the flow visualization figures, and also have split figure 9 into two separate figures, with 9A-9H now appearing as figure 9, and 9I-9P appearing as figure 10 in the revised version of the manuscript.

We have also calculated the divergence of velocity for the velocity fields shown in figures 7-10, and added discussion on the meaning of non-zero 2D divergence in an incompressible flow (out-of-plane motion), and identified the most likely source of this divergence.

For more detailed explanation of the revisions, please see the responses to reviewers below.

We would like to thank both the reviewers for their comments, which have helped in strengthening our manuscript.

Reviewer: 1

Comments to the Author(s)

The effects of varying kinematics of metachronal wave paddling on the resulting flow are nicely illustrated in this manuscript.

I enjoyed reading it and think it an important contribution to our knowledge of this method of locomotion which is so widespread throughout the animal kingdom.

The paper is generally clear and well written. The results of the experiments with the robotic model confirm what experts would have expected and I expect it to enter the textbooks on animal locomotion from now on.

1) The figures showing the flow visualization results can be made much clearer. For example, by decreasing the number of arrows and increasing the size of the figures (figures 3 – 9 and 10A).

We have revised figures 3-9 to have lower vector density and have increased the size of part figures.

We have split up figure 9 into 2 figures, in the revised manuscript (Figure 9 and Figure 10).

We have reduced vector density in figure 10A of the original submission without increasing figure size, so that it will fit in 1 figure when combined with the other parts of Figure 10. Please refer to Figure 11A in the revised manuscript.

2) Hinge kinematics including the limits to the hinge angles should be given in the experimental methods section (2).

We have moved the entire subsection 3.1 on hinge kinematics to a new subsection 2.2 in the revised manuscript, as shown below.

Lines 131-145:

2.2 Paddling kinematics

Kinematics achieved by the 4 paddles were tracked from raw PIV images using the image analysis program ImageJ [25]. The paddle angles ($\alpha(t)$, see Figure 2) were measured at 10 points in a cycle. The definition of α was identical to that used in a study of krill swimming kinematics by Murphy et al. [8]. α increases in PS (cycle fraction from 0-0.5) and decreases in RS (cycle fraction from 0.5-1). The results show that achieved paddle angles (markers in Figure 2) closely follow the kinematics prescribed to the stepper motors (solid lines in Figure 2) across all Re and phase lag conditions.

The hinges on the paddles were allowed to rotate freely about angle β (defined in Figure 2A, same as in [8]). β varied between a minimum angle of approximately 120° and a maximum angle of 180° . Variation of β angles are provided as electronic supplementary material (Figure S1-Figure S3). β reaches its maximum value for each paddle at the beginning of their respective PS as α accelerates during stroke duration from 0 to 0.25; and begins decreasing once α begins to decelerate during stroke duration from 0.25 to 0.5. The rate of decrease of β increases with increasing Re, and is also delayed in time. In general, β spends more time near its peak value than its minimum, which agrees with observations in hovering *E. superba* [8].

Reviewer: 2

Comments to the Author(s)

1) My main concern is that, while the experiments are designed to produce 2D flow fields, the plots show regions with arrows converging or diverging to a point (e.g. to the right of the paddles in Fig 3A, 3B, etc.), implying 3D flow. Could the authors quantify this (e.g. by computing the divergence in 2D) and discuss possible reasons why the flow is going in or out of the laser sheet?

We calculated the divergence of the 2D velocity vector fields in Figure 7-Figure 10 using the following equation:

$$\nabla \cdot \vec{U} = \frac{\partial u}{\partial x} + \frac{\partial v}{\partial y}$$

The results are shown below and are also included in supplementary material Figures S5-S8 of the revised manuscript. We observed non-zero divergence in the flow field, particularly in the region behind the paddles, and between adjacent paddles when they were moving away from each other, where the velocity vector field resembled that of a point source or a point sink. Though our paddles were idealized as 2D flat plates, there was a gap of 3.4 cm on either edge (z-dimension) that could permit three-dimensional fluid motion in/out of the laser sheet. This gap was needed to fit the driving shafts on the paddle roots. A previous study of drag-based paddling by Kim and Gharib (2011) showed that 3D tip vortices are formed behind a rectangular flat plate. We expect that the gaps promoted formation of 3D flow features that likely resulted in regional artifacts in the 2D flow field that looked like a point source/point sink. This has been discussed in section 3.2 of the revised manuscript, as shown below:

Lines 333-353:

Divergence of the 2D velocity fields, in the form of point sources (vectors pointing away from each other) or point sinks (vectors pointing towards each other), can be seen in some instances in the flow fields (Figures 3-10). Flow sources and sinks indicate fluid motion was out of the plane where PIV recordings were obtained. The extent of three-dimensionality of the flow was characterized by computing 2D divergence of the velocity field, defined in equation (18) below:

$$\nabla \cdot \vec{U} = \frac{\partial u}{\partial x} + \frac{\partial v}{\partial y} \quad (18)$$

A region with positive divergence appears in the velocity field as a point source, while a region with negative divergence appears as a point sink. Divergence was calculated for the velocity fields shown in Figure 7-Figure 10 (0% phase lag at Re=250, and 25% phase lag at Re = 250, 50, and 800, respectively), and are provided as electronic supplementary material (Figures S5-S8). Regions of positive divergence were formed behind the tail-most paddle at the end of synchronous recovery (Figure S5), as well as between paddles moving away from each other during metachronal motion (Figure S6). Regardless of Re, positive divergence regions were generated in the same regions for each case, but were of much smaller magnitude at Re=50 (Figure S7) and higher magnitude at Re=800 (Figure S8), when compared to the divergence at Re=250 (Figure S6). Though our paddles were idealized as 2D flat plates, the mechanical system used to drive paddle motion required gaps between the edges of the paddles and the sides of the tank (along the z-axis). These gaps likely allowed for fluid to be entrained from out of the rotational plane of the paddles, resulting in some three-dimensionality of the flow. This is supported by a previous study by Kim and Gharib [26], where rotational motion of a single flat-plate paddle was shown to result in the generation of a three-dimensional tip vortex that entrains flow from outside of the rotational plane (along the axis of rotation).

Discussion of the gaps on the sides of the paddles was also added to section 2.1 and the caption of figure 1 in the revised manuscript, as shown below:

Lines 101-106:

Rotational motion was controlled using timing belts that were driven by 2-phase hybrid stepper motors with integrated encoders (ST234E, National Instruments Corporation, Austin, TX, USA), with 20,000 steps per revolution resolution. The 6.35 mm diameter aluminum shafts protruded from either side of the 3D printed base, and were driven by the rotation of the timing belts. In order to mount timing belt pulleys to the shafts, 3.4 cm gaps were left between the paddles and the walls on either side of the robotic platform.

Lines 586-593 (caption of Figure 1B):

(B) Top view of the paddling model showing the location of PIV camera. Laser sheet for PIV was located at the central plane of the paddles. Aluminum shafts (22 cm long, 6.35 mm diameter) protruding from either side of the model (top and bottom of B) were coupled to stepper motors with timing belts in order to drive each paddle with independently prescribed motion profiles. In order to mount timing belt pulleys to the shafts, 3.4 cm gaps were left between the paddles and the walls on either side of the robotic platform. These gaps have little impact on the flow at Re=50, but do contribute to some out-of-plane flow at Re=250 and Re=800. A false acrylic wall (thickness=0.6 cm) was used to reduce the width of the tank from 29 cm to 22 cm.

Figure S5

Figure S6

Figure S7

Figure S8

2) In addition, it was not always clear which specific vortices and jets the main text referred to in each of the figure panels. It would be helpful if the key regions or set of arrows could be somehow highlighted or distinguished, e.g. with the use of additional markers or pointers.

We have labeled the vortices generated by the right paddle during synchronous and metachronal paddling in Figure 3-Figure 4.

We have labeled the large-scale downward jet generated by metachronal paddling in Figure 8.

Additionally, the text in sections 3.1 and 3.2 were revised to provide more clarity on development of tip vortices in the 2-paddle system, and generation of the large-scale wake in the 4-paddle system (see below):

Lines 251-278:

Flow generated during PS by synchronous, periodic motion of two paddles consists of co-rotating vortices near the tip of each paddle (Figure 3A-Figure 3D). During the first half of PS, shear layers form near the tip of each paddle (Figure 3B), which then roll-up into negatively signed (clockwise) vortices (Figure 3C). These co-rotating vortices are shed from the paddles near the end of PS (Figure 3D), and propagate below the paddles with further progression of the stroke cycle. Viscous dissipation of these vortices occurs subsequently, as evidenced by decreasing vorticity magnitude of the negatively-signed vortices in Figure 3D-Figure 3H. Similar to PS, an oppositely-signed (counterclockwise) vortex is generated from the tip of each paddle during RS (Figure 3F). The interaction of the counterclockwise pair of vortices in RS with the previously shed clockwise vortices from PS (Figure 3G) results in generating a jet primarily in the horizontal orientation (Figure 3H). Vortex formation and propagation during PS and RS of the right paddle are indicated by dashed red boxes in Figure 3.

Metachronal paddling allows for the formation of counter-rotating vortices at the tips of adjacent paddles. The interaction of these counter-rotating vortices formed during metachronal paddling at a phase lag of 16.7% results in the formation of an angled jet moving away from the body (Figure 4). The right paddle leads the PS, while the left paddle is phase-delayed in starting its PS. The shear layer formed by PS of the right paddle rolls up into a clockwise vortex that detaches from the tip near the end of PS (Figure 4C). However, the left paddle does not start PS until 60% PS of the right paddle (Figure 4C). As a result of the phase delay, the clockwise vortex formed at the tip of the right paddle is stronger in magnitude compared to the co-rotating (clockwise) vortex formed at the tip of the left paddle. The shed vortex from the right paddle at the end of its PS (Figure 4D) tailors the flow more downward when compared to the same phase point in the synchronous case (Figure 3D). RS of the right paddle at 16.7% phase lag generates a shear layer with oppositely signed vorticity compared to that of the left paddle, which completes its PS during the RS of the right paddle (Figure 4E-Figure 4F). The interaction of the counter-rotating vortices shed from both paddle tips occurs at 60% RS of the right paddle (Figure 4G), generating a bulk flow that moves downward in the head-to-tail direction (Figure 4H). Vortex formation and propagation during PS and RS of the right paddle are indicated by dashed red boxes in Figure 4.

Lines 303-305:

For $Re=250$ at 25% phase lag (Figure 8), a series of small-scale jets form as each pair of paddles come near each other, merging to sustain a continuous large-scale vertical jet (indicated by a dashed red box in Figure 8A).

Figure 3

Figure 4

Figure 8